# Apo and Aβ46-bound γ-secretase structures provide insights into amyloid-β processing by the APH-1B isoform

Ivica Odorčić ®[1,2,3,4], Mohamed Belal Hamed ®[3,4], Sam Lismont[3,4], Lucía Chávez-Gutiérrez[3,4,5] ✉ & Rouslan G. Efremov ®[1,2,5] ✉

Deposition of amyloid-β (Aβ) peptides in the brain is a hallmark of Alzheimer's disease. Aβs are generated through sequential proteolysis of the amyloid precursor protein by the γ-secretase complexes (GSECs). Aβ peptide length, modulated by the Presenilin (PSEN) and APH-1 subunits of GSEC, is critical for Alzheimer's pathogenesis. Despite high relevance, mechanistic understanding of the proteolysis of Aβ, and its modulation by APH-1, remain incomplete. Here, we report cryo-EM structures of human GSEC (PSEN1/APH-1B) reconstituted into lipid nanodiscs in apo form and in complex with the intermediate Aβ46 substrate without cross-linking. We find that three non-conserved and structurally divergent APH-1 regions establish contacts with PSEN1, and that substrate-binding induces concerted rearrangements in one of the identified PSEN1/APH-1 interfaces, providing structural basis for APH-1 allosteric-like effects. In addition, the GSEC-Aβ46 structure reveals an interaction between Aβ46 and loop 1[PSEN1], and identifies three other H-bonding interactions that, according to functional validation, are required for substrate recognition and efficient sequential catalysis.

Alzheimer's disease (AD), the most common form of dementia, begins with the accumulation of amyloid-β (Aβ) peptides in the brain 2-3 decades before symptoms manifest[1]. Aβ peptides of different lengths, ranging between 37 and 43 amino acids (aa), are generated by sequential cleavage of the amyloid precursor protein (APP) by the γ-secretase complexes (GSECs)[2,3]. However, it is the cerebral accumulation of longer, aggregation-prone, Aβ peptides (≥42 aa in length) which triggers toxic molecular and cellular cascades that lead to neuronal loss[4].

GSECs are intramembrane multimeric proteases that cleave numerous type-I transmembrane proteins with short ectodomains and no sequence homology[5]. The ample substrate repertoire implicates GSEC activity in several biological pathways; the best characterised being Notch and APP. Processing of Notch is essential in organism development and dysregulated in cancer[6], whereas APP processing is associated with AD pathogenesis.

GSECs are composed of presenilin (PSEN, the catalytic subunit), nicastrin (NCT), APH-1 and PEN-2[7,8]. The assembly of a tetrameric proenzyme triggers PSEN autoproteolysis and generates a pentameric active complex in which the catalytic site is formed at the interface between PSEN N- and C-terminal fragments (NTF and CTF, respectively; Fig. 1a). In humans, the presence of two isoforms of PSEN (PSEN1, PSEN2) and two of APH-1 (APH-1A, APH-1B) generates a family of four highly homologous GSEC complexes[9,10] that are distinguished by different subcellular localisations and particular kinetics of substrate proteolysis[11,12]. GSECs containing PSEN2 and/or APH-1B subunits produce a larger proportion of longer and aggregation-prone Aβ peptides, relative to those with PSEN1 and/or APH-1A subunits

[1]Center for Structural Biology, VIB, Brussels, Belgium. [2]Structural Biology Brussels, Department of Bioengineering Sciences, Vrije Universiteit Brussel, Brussels, Belgium. [3]VIB-KU Leuven Center for Brain & Disease Research, Herestraat 49 box 602, 3000 Leuven, Belgium. [4]Department of Neurosciences, Leuven Brain Institute, KU Leuven, Herestraat 49 box 602, 3000 Leuven, Belgium. [5]These authors jointly supervised this work: Lucía Chávez-Gutiérrez, Rouslan G. Efremov. ✉e-mail: Lucia.ChavezGutierrez@kuleuven.be; Rouslan.Efremov@vub.be

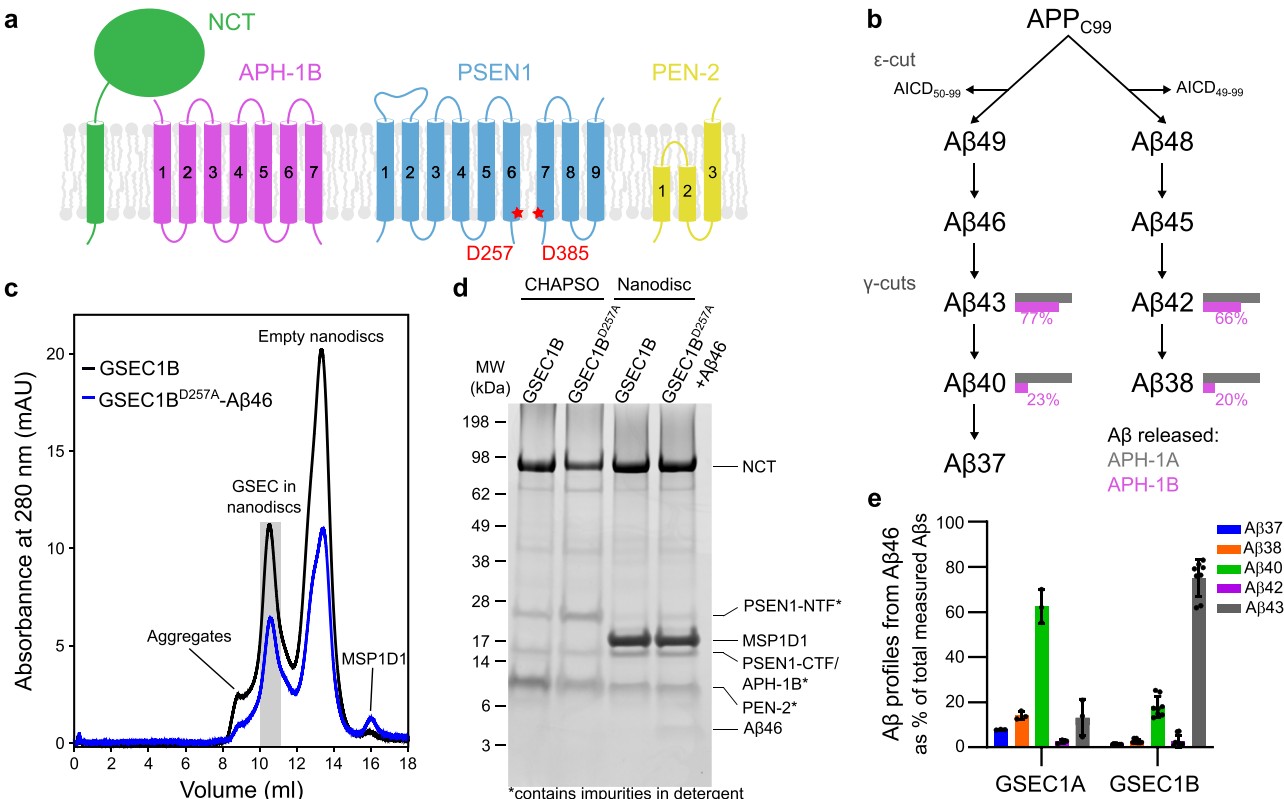

**Fig. 1 | Reconstitution of GSEC1B into lipid nanodiscs. a** Schematic representation of GSEC subunits. The catalytic aspartates are indicated, and their respective positions are marked with red stars. **b** Sequential processing of APP$_{C99}$ by GSEC and the degree of processivity (%) between the APH-1A and APH-1B isoforms in detergent conditions are indicated in grey and pink, respectively. **c** Size exclusion chromatograms of GSEC1B and GSEC1B$^{D257A}$-Aβ46 after reconstitution into MSP1D1 lipid nanodiscs. Grey area shows peak fraction used for cryo-EM. **d** Coomassie stained SDS-PAGE of purified GSEC1B (WT and D257A mutant) solubilised in CHAPSO and reconstituted into lipid nanodiscs. Aβ46 was added to the purified GSEC1B$^{D257A}$ prior to reconstitution. **e** ELISA-based quantification of de novo Aβ products generated from Aβ46 by GSEC1A or GSEC1B reconstituted into lipid nanodiscs. Aβ profiles show the percentage of Aβ products, relative to total measured Aβ (37, 38, 40, 42 and 43) peptides. Data are presented as mean ± SD, n = 3 for GSEC1A and n = 8 for GSEC1B. The amounts of Aβ products measured are shown in Supplementary Fig. 1c. Source data are provided as a Source Data file.

(Fig. 1b)[11]. Consistently, genetic inactivation of the mouse *APH-1B* gene in an AD mouse model reduces disease–relevant phenotypic features[13]. Recently, genetic studies identified the *APH-1B* gene as predisposing to AD[14].

Cleavage of APP by the β- or α-secretases[15,16] removes its large ectodomain and generates a transmembrane C-terminal fragment either 99 or 83 aa in length (APP$_{C99}$ or APP$_{C83}$), respectively. These fragments are then proteolysed sequentially within their transmembrane (TM) domains by GSEC: a first endopeptidase-like cut at position 48 or 49 releases the APP intracellular domain (AICD). Next, the resulting substrate is proteolysed by three to four residues at a time by several sequential carboxypeptidase-like cuts[2,3,17] (Fig. 1b). Every cut lowers the stability of the successive enzyme-substrate (E-S) complexes[18] until Aβ40/Aβ37, or Aβ42/Aβ38 in the other product-line, are released to the extracellular/luminal environment. Mutations in PSEN that destabilise GSEC-APP/Aβ interactions enhance production of the longer and more hydrophobic Aβ42 and Aβ43 peptides[18] and cause early-onset familial AD (FAD)[19].

Recent cryo-EM structures of GSEC (PSEN1/APH-1A) in complex with APP$_{C83}$[20] or Notch[21] substrates revealed that E-S complex formation is associated with substantial conformational rearrangements in both enzyme and substrate to reach remarkably similar structures of the complexes, despite the low sequence homology between these substrates. In both structures, the transmembrane region of the substrate unwinds close to the scissile bond and forms a hybrid substrate-GSEC β-sheet structure.

Despite the wealth of structural information on GSEC, the mechanistic bases of GSEC processivity (sequential cleavage) and its modulation by isoforms of the APH-1 subunit remain poorly characterised. Importantly, enzyme processivity is regulated at the level of GSEC-Aβ$_n$ interactions; however, only the structures of GSEC with the initial endopeptidase APP/Notch substrates are known.

In this work we report the cryo-EM structures of GSEC (PSEN1/APH-1B, referred to as GSEC1B) complexes in apo form and in complex with the intermediate Aβ46 substrate in a native-like environment and in the absence of E-S cross-linking. Structural comparison with GSEC1A (PSEN1/APH-1A) shows concerted isoform-dependent structural changes at the active site (PAL motif; PSEN1 Pro433-Leu435) and at the PSEN1/APH-1 interface, upon substrate-binding. This provides structural understanding of the involvement of the APH-1 subunit in substrate gating/processing in an isoform-dependent manner. The GSEC-Aβ46 structure shows conservation of H-bonding interactions between GSEC and the initial or intermediate substrates, while functional studies establish their contribution to substrate stabilisation and processing. Taken together, these findings indicate that the substrate backbone structure is remarkably preserved during sequential proteolysis.

## Results

### Cryo-EM structure determination
We characterised the GSEC1B isoform in apo form and in complex with the Aβ46 peptide using cryo-EM single particle analysis. Aβ46 is an

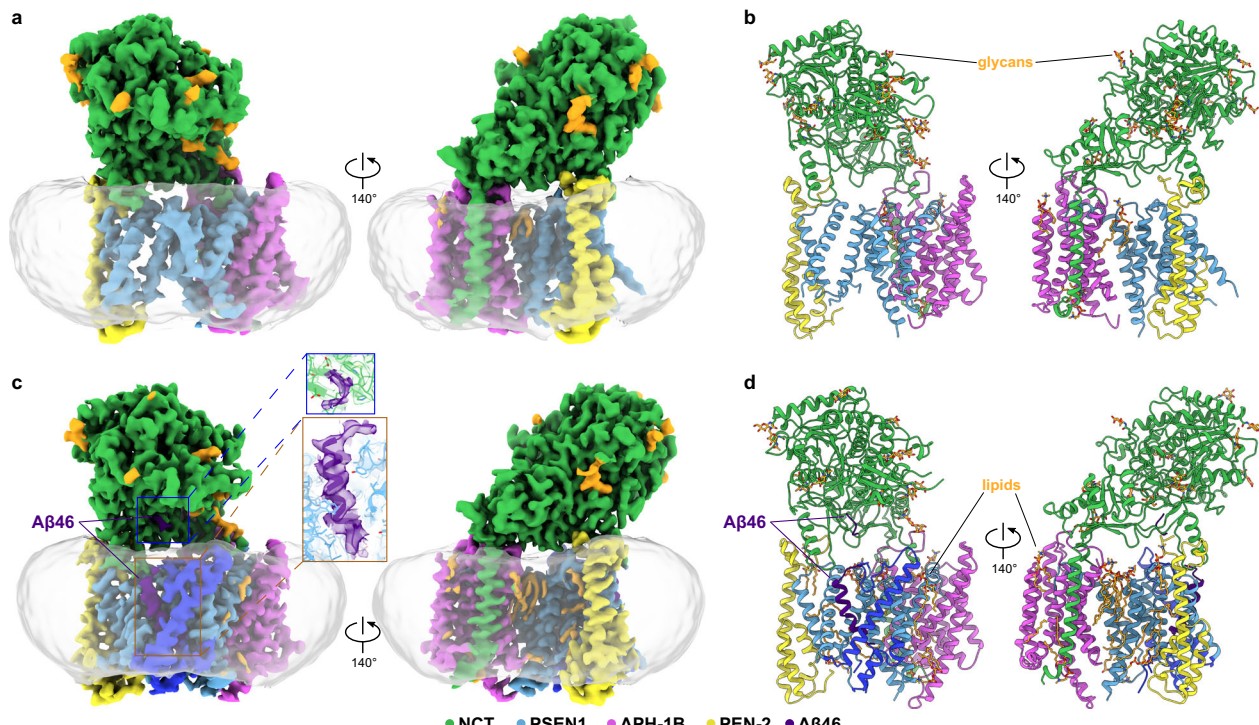

● NCT   ● PSEN1   ● APH-1B   ● PEN-2   ● Aβ46

**Fig. 2 | Cryo-EM structures of GSEC1B and GSEC1B-Aβ46 complex in lipid nanodiscs. a** Cryo-EM map of apo GSEC1B coloured by subunit. Resolved glycans in NCT subunit and density corresponding to ordered lipids are coloured in orange. The density corresponding to lipid nanodisc extends ~ 2 nm around the edge of GSEC and is ~ 4 nm thick, with the thickest part found next to PEN-2, and the thinnest part close to APH-1B. **b** Atomic model of apo GSEC1B. **c** Cryo-EM density map of GSEC1B-Aβ46 complex, Aβ46 shown in purple. Regions of PSEN1 resolved in the complex with Aβ46 but not in apo state are shown in dark blue. Density of Aβ46 N-terminus proximal to Glu650^NCT and of the density of Aβ46 TM domain are shown in the inset. The maps shown in panels **a** and **c** were filtered using Gaussian filter for better visualisation. **d** Atomic model of the GSEC1B-Aβ46 complex.

intermediate product generated during the sequential (carboxypeptidase-like) cleavage of APP and is hydrolysed to the major Aβ40 product (Fig. 1b). We purified human GSEC1B from High Five insect cells[11] and reconstituted it into lipid nanodiscs to more closely mimic the membrane environment[22] (Fig. 1c, d and Supplementary Fig. 1a, b). Screening for reconstitution conditions established that the combination of membrane scaffold protein (MSP) MSP1D1 and a 1:2 (M/M) POPC:DLPC lipid mixture produced homogeneous GSEC nanodiscs suitable for high-resolution cryo-EM (Supplementary Figs. 2–5).

Previous structural analyses of GSEC in complex with its substrates have generated high-resolution structures with the initial endopeptidase substrates APP_C83[20] and Notch[21]. To gain insights into GSEC-substrate interactions during carboxypeptidase-like cleavages, we generated an inactive GSEC1B enzyme, containing a catalytically inactive but mature PSEN1 D257A mutant (pentameric GSEC1B^D257A), in complex with Aβ46 (Fig. 1c, d). A similar strategy, using the inactive GSEC1A^D235A (with PSEN1 D385A) with the addition of a covalent cross-linking bond to stabilise E-S complex was used in the above-mentioned structures.

Incubation of Aβ46 with the reconstituted wild-type (WT) GSEC1A and GSEC1B complexes resulted in generation of shorter Aβ products, including Aβ43, Aβ40 and Aβ37 as well as Aβ42 and Aβ38 (Fig. 1e and Supplementary Fig. 1c). We note that Aβ46 is mainly converted into Aβ43, Aβ40 and Aβ37, but its processing to Aβ42 and Aβ38 also occurs albeit to a lesser extent[2]. Strikingly, de novo Aβ production revealed that the processivity difference between the isoforms in nanodisc conditions is even greater than previously reported for detergent conditions[11]. The reconstituted inactive GSEC1B^PSEN1 D257A mutant did not hydrolyse Aβ46, however, it formed stable E-S complexes in the absence of covalent cross-linking (see below).

The inspection of cryo-EM samples of GSEC1B on holey grids revealed that more than 90% of particles were fragmented, likely

because of interaction with the air-water interface[23,24]. This difficulty was overcome by using high-coverage graphene oxide-coated grids, utilising a refined protocol relying on poly-L-lysine pre-coating[25] (see Materials and Methods). To improve particle orientation distribution, the graphene oxide surface was coated with polyethylene glycol (PEG). Similar conditions were applied to reconstruct the structure of the GSEC1B^D257A-Aβ46 complex (Supplementary Figs. 3 and 5).

## Structure of apo GSEC1B

We solved the structure of GSEC1B in lipid nanodiscs to an overall resolution of 3.3 Å (Fig. 2a, b, Table 1, and Supplementary Figs. 2, 3). The map was of sufficient quality to model 68% (or 1224 residues) of the structure (Supplementary Table 1). The subunits NCT, APH-1B and PEN-2 were modelled almost entirely, whereas around half of the encoded polypeptide was modelled for PSEN1. Specifically, loop 1 (residues 103-124, connecting TM1-TM2), TM2, part of TM6, and the large intracellular loop between TM6-TM7, except for a 9-residue stretch preceding the autoproteolytic cleavage site (Glu280-Ser289), were not resolved in PSEN1. Three elongated densities in the membrane-embedded region at the interfaces between TM1,8^PSEN1 and TM4^APH-1B; TM1,4,7^APH-1B and NCT^TM; and TM5,7^APH-1B were modelled as phospholipids (Fig. 2a, b). The latter replaces a cholesterol moiety reported in digitonin solubilised GSEC1A structures[20,21,26,27].

Structural comparison of GSEC1B with the apo state model of human GSEC1A solved in amphipols[28,29] showed a high overall similarity (RMSD of 1.2 Å over 7918 atoms; Fig. 3a, Supplementary Table 2 and Supplementary Movie 1). Nevertheless, the transmembrane region of GSEC1B was expanded by ~ 2 Å as compared to GSEC1A (Supplementary Movie 1).

To gain insights into the APH-1 isoform-dependent allosteric-like effects, we analysed the structural differences between APH-1A and

**Table 1 | Cryo-EM data collection, refinement, and validation statistics**

| | #1 GSEC1B apo (EMDB-17112) (PDB 8OQY) | #2 GSEC1B-Aβ46 complex (EMDB-17113) (PDB 8OYZ) |
|---|---|---|
| **Data collection and processing** | | |
| Magnification | 60,000 | 60,000 |
| Voltage (kV) | 300 | 300 |
| Electron exposure (e–/Å²) | 63.6 | 56.2 |
| Defocus range (μm) | −0.5 to −3.5 | −0.5 to −3.0 |
| Pixel size (Å) | 0.776 | 0.759 |
| Symmetry imposed | C1 | C1 |
| Initial particle images (no.) | 986,830 | 2,433,778 |
| Final particle images (no.) | 115,197 | 53,612 |
| Map resolution (Å) | 3.3 | 3.4 |
| FSC threshold | 0.143 | 0.143 |
| Map resolution range (Å) | 3.2 to 5.5 | 3.3 to 5.0 |
| **Refinement** | | |
| Initial model used (PDB code) | 5FN5 | 6IYC |
| Model resolution (Å) | 3.3 | 3.4 |
| FSC threshold | 0.5 | 0.5 |
| Map sharpening B factor (Å²) | −95.8 | −80.4 |
| **Model composition** | | |
| Non-hydrogen atoms | 9810 | 11131 |
| Protein residues | 1224 | 1340 |
| Ligands | 20 | 28 |
| **B factors (Å²)** | | |
| Protein | 55.10 | 49.00 |
| Ligand | 45.11 | 42.18 |
| **R.m.s. deviations** | | |
| Bond lengths (Å) | 0.005 | 0.004 |
| Bond angles (°) | 0.923 | 0.849 |
| **Validation** | | |
| MolProbity score | 1.55 | 1.63 |
| Clashscore | 4.44 | 6.40 |
| Poor rotamers (%) | 0.71 | 0.27 |
| **Ramachandran plot** | | |
| Favoured (%) | 95.28 | 95.92 |
| Allowed (%) | 4.55 | 3.85 |
| Disallowed (%) | 0.17 | 0.23 |

APH-1B subunits. In humans, subunits APH-1A and APH-1B share a 56% sequence identity with mostly conservative substitutions scattered throughout their sequence (Fig. 3b, e). Despite the very similar overall structures (RMSD of 0.9 Å over 1283 atoms) the backbones of the isoforms diverged locally at three segments on the intracellular (TM3-4) and extracellular (TM2-3 and TM6-7) membrane surfaces, with changes mapping to the PSEN1/APH-1 interface (Fig. 3a, b, red dashed boxes). Specifically, the extracellular ends of TM2-TM3 and TM6-TM7 helical pairs were bent by 4° and 7°, respectively; whereas the cytosolic TM3-TM4 connecting loop (residues 104-110) was partially disordered in APH-1B but resolved in APH-1A. Notably, these structural differences coincide with local clusters of sequence divergence between the APH-1 isoforms (Fig. 3e).

To get further insight into a possible APH-1 isoform-driven allosteric-like mechanism, we examined the PSEN1/APH-1 interface.

This interface spans an area of ~ 2000 Å² and involves the interaction between adjacent α-helices of APH-1 (TM2-TM4) and PSEN1 (TM1, TM8-TM9) as well as the insertion of the PSEN1 C-terminus into the APH-1 helical bundle on the extracellular side (Fig. 3c). The interactions between PSEN1 and APH-1 are conserved between the isoforms, except for four substitutions in the transmembrane helices (APH-1B/APH-1A: I32/V32, I36/V36, L51/V51 and M127/I128; Fig. 3d) and several substitutions in the pocket where the PSEN1 C-terminus binds (APH-1B/APH-1A: N62R, K69Y, T136I, F155Y, Y159T, M162L, V199T, S206N; Supplementary Table 3). We speculate that despite the overall structural similarity, differences in length, charge and/or polarity of the side chains forming the PSEN1/APH-1 interface (Supplementary Table 3) might alter enzyme dynamics or its structure when the substrate binds, thus leading to the observed functional effect.

Significant conformational differences between GSEC1A and GSEC1B structures in apo states are observed in the catalytic PSEN1 subunit (Supplementary Fig. 6). In the active site of PSEN1, the TM8-TM9[PSEN1] loop containing the conserved and functionally important PAL motif[30] is resolved in GSEC1B, but not in GSEC1A. The cytoplasmic end of TM8[PSEN1] interacts with APH-1A but bends away from APH-1B by ~16°, suggesting that the difference in its dynamics might be linked with APH-1 isoforms. Moreover, the cytoplasmic end of TM6[PSEN1] (between the catalytic Asp257 and Pro264) and the first helical turn of TM7[PSEN1] (Val379-Gly382) were structured in GSEC1A but unresolved in GSEC1B. In the latter, the volume of the ordered cytoplasmic end of TM6[PSEN1] (Leu258-Pro264) in GSEC1A is occupied by the TM8-TM9[PSEN1] loop. In addition, the PSEN1 Glu280-Ser289 fragment of the long intracellular TM6-TM7 loop (residues 278-382) is resolved at the interface between PSEN1 TM3 and TM7 only in GSEC1B structure (Fig. 3a) where it occupies the same position as in the substrate-bound GSEC1A structures[20,21]. Lastly, the cytoplasmic end of TM3 is extended by two helical turns (Lys160-Leu166) in GSEC1B, relative to GSEC1A (Fig. 3a). The observed conformational differences in the PSEN1 subunit may arise from the interaction with different APH-1 isoforms and/or from the different lipid mimetic environments (nanodiscs versus amphipols) used to solve the structures of the complexes.

### Structure of GSEC1B-Aβ46 complex
We next investigated GSEC-Aβ interactions and the role of APH-1 isoforms in substrate processing by determining the GSEC1B structure in complex with Aβ46. For structural analysis, the inactive GSEC1B[D257A] mutant was reconstituted into lipid nanodiscs in the presence of Aβ46 (Fig. 1c, d) and supplemented with an excess of Aβ46 prior to plunge-freezing the cryo-EM grids. The reconstituted GSEC1B-Aβ46 complex was sufficiently stable for structural analysis without the need for cross-linking, a strategy previously used for stabilisation of GSEC-substrate complexes[20,21].

Extensive 3D classification, after partial signal subtraction, enabled the isolation of a uniform population of GSEC1B-Aβ46 complexes and their reconstruction to a resolution of 3.4 Å (Fig. 2c, d, Table 1, and Supplementary Figs. 4, 5). In the presence of substrate, parts of PSEN1 disordered in apo structure became resolved; we modelled 73% of the GSEC1B sequence (Supplementary Table 1), including a 24-residue-long continuous Aβ46 fragment. A total of 14 boundary phospholipids were also modelled in the map (Fig. 2c, d).

Aβ46 binding triggered substantial structural rearrangements within the catalytic subunit (Fig. 4 and Supplementary Movie 2) that are resembling those previously reported[20,21]. The overall structure is remarkably similar to the GSEC1A in complex with Notch, APP[C83] or the transition state analogue inhibitor L-685,458, which induces a substrate-bound-like conformation. In particular, the structure of the catalytic dyad in PSEN1[D257A] of GSEC1B-Aβ46, PSEN1[D385A] of GSEC1A-APP[C83], and PSEN1[WT] of GSEC1A-Inhibitor (L-685,458) is virtually identical (Supplementary Fig. 7, Supplementary Table 4) indicating that

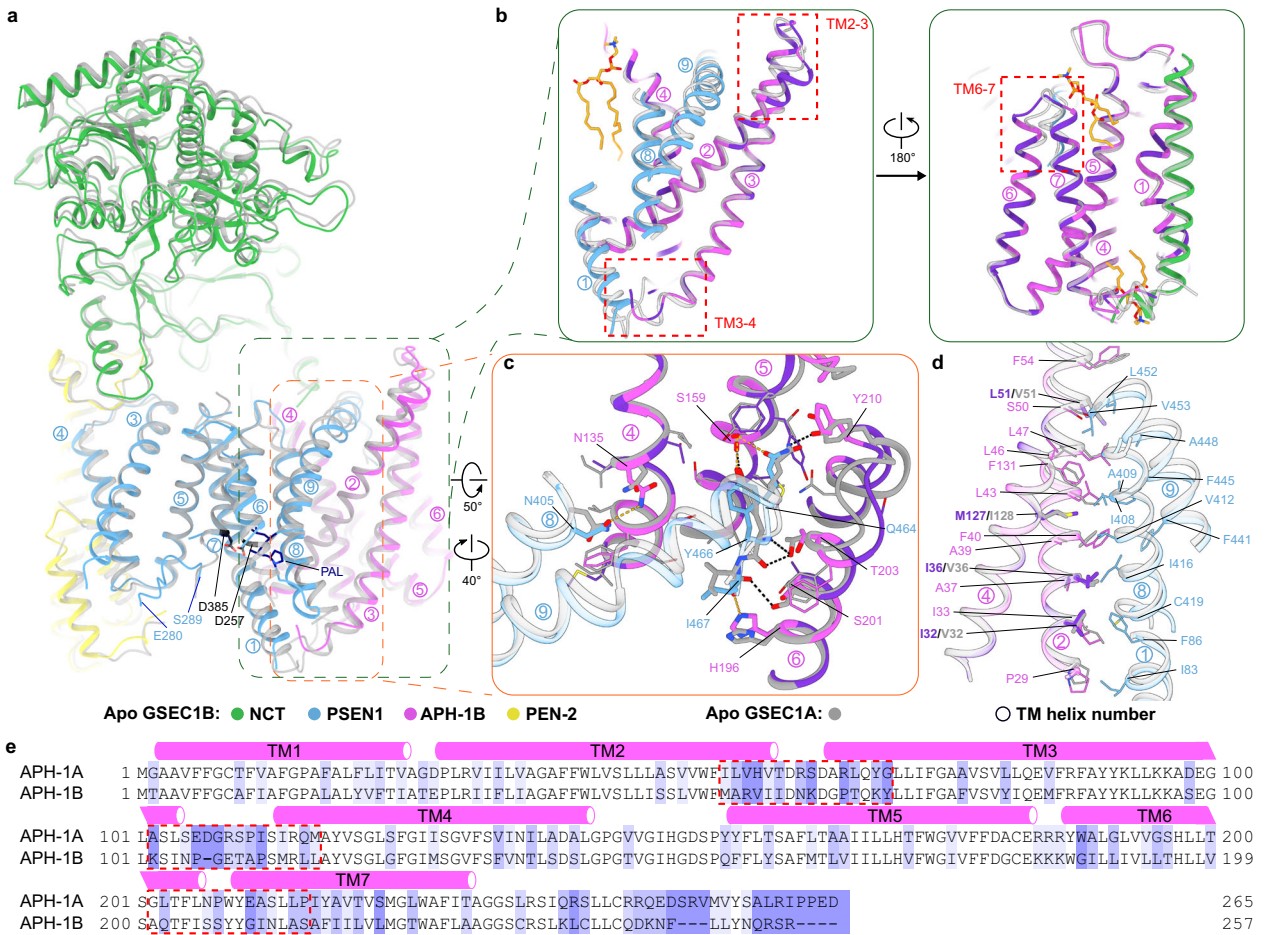

**Fig. 3 | Structural differences at the PSEN1/APH-1 interface. a** Structural comparison of GSEC1B and GSEC1A (PDB: 5FN5). GSEC1B is colour-coded as in Fig. 2, GSEC1A is shown in grey. PSEN1 and APH-1 TMs are indicated with colour-coded circled numbers and the sidechains of the catalytic aspartates and the PAL motif are shown as sticks. The segment indicated in the orange box is shown in detail in (**c**). **b** Structural differences between APH-1 isoforms. Residues that differ between the isoforms are coloured in purple. The red dashed boxes indicate the three structurally different regions of APH-1. **c** Details of the interfaces between C-terminus of

PSEN1 and APH-1 isoforms. Putative hydrogen bonds between PSEN1 and APH-1B are shown in orange, and hydrogen bonds between PSEN1 and APH-1A are shown in black. **d** Details of the transmembrane PSEN1/APH-1 interface. **e** Sequence alignment of human APH-1A and APH-1B. Positions of TM helices are indicated graphically. The red dashed boxes indicate structurally different regions in APH-1 as shown in (**b**). Poorly conserved sequence regions are shaded in blue according to the degree of divergency.

neither D257A nor D385A mutations alter the active site structure. Nevertheless, specific structural differences between the initial endopeptidase substrates and intermediate Aβ46 are observed in functionally relevant regions in PSEN1 (Fig. 5a–c) (see below).

The PSEN1 subunit has high similarity (RMSD of 0.6 Å) in all three (Aβ46/APP$_{C83}$/Notch[20,21]) substrate-bound structures (Supplementary Table 2). When compared to the apo state, the changes in PSEN1 include: ordering of TM2, adjacent loop 1 and intracellular extensions of TM6 and TM7 (Fig. 2c, d and Fig. 4; dark blue); bending of TM1; shifts of TM3 and TM6 by ~5 Å, relative to the substrate; and shift of the PAL motif containing loop by 3.4 Å toward the substrate (Fig. 4 and Supplementary Movie 2).

Our structures allowed a direct comparison of apo versus substrate-bound states in the same, native-like, environment. They showed that substrate-induced conformational changes in PSEN1 propagate into a divergent region in the APH-1 isoforms (TM3-TM4 loop) (Fig. 4b and Supplementary Fig. 8a). An apparent allosteric-like pathway links TM8-9$^{PSEN1}$ loop with the extracellular end of TM1$^{PSEN1}$ which in turn interacts with the TM3-TM4$^{APH-1}$ loop. The sequence, structures, length and charges of TM3-TM4$^{APH-1}$ loop are different between GSEC1A and GSEC1B with bound substrate (Fig. 3e, Supplementary Fig. 8a).

The conformations of TM1$^{PSEN1}$, which links the (active site) TM8-TM9$^{PSEN1}$ loop with APH-1, are similar between the substrate-bound structures of GSEC1A and GSEC1B suggesting that the structural differences alone cannot account for the observed allosteric-like effect on APH-1 isoforms. However, TM1$^{PSEN1}$ is highly dynamic. Upon substrate binding, it shifts vertically by ~ 1.5 Å towards the extracellular side, supporting the notion that piston-like movement of TM1$^{PSEN1}$ is associated with activity[31,32]. Furthermore, its intracellular end bends by 33° towards APH-1 at Pro88 which functions as a hinge. Interestingly, the FAD-causing P88L mutation strongly impairs processivity and causes release of Aβ45 and longer Aβ species in vitro[33,34]. Because Leu, a helix-promoting residue, removes the hinge property enabled by Pro, a kink-forming residue, the effects of the pathogenic mutation suggest that the described motion of TM1$^{PSEN1}$ is necessary for sequential catalysis.

The density of the Aβ46 substrate was well-resolved for the backbone (Fig. 6b, Supplementary Fig. 4), but did not allow the assignment of sequence register with confidence, hence, we modelled it as a polyalanine chain. We have attempted local 3D classification as well as employed various heterogeneity analyses on the substrate and the surrounding area, but nothing yielded a higher resolution reconstruction. We note that the sequence of the membrane-embedded fragment of Aβ46 does not contain bulky aromatic residues but does

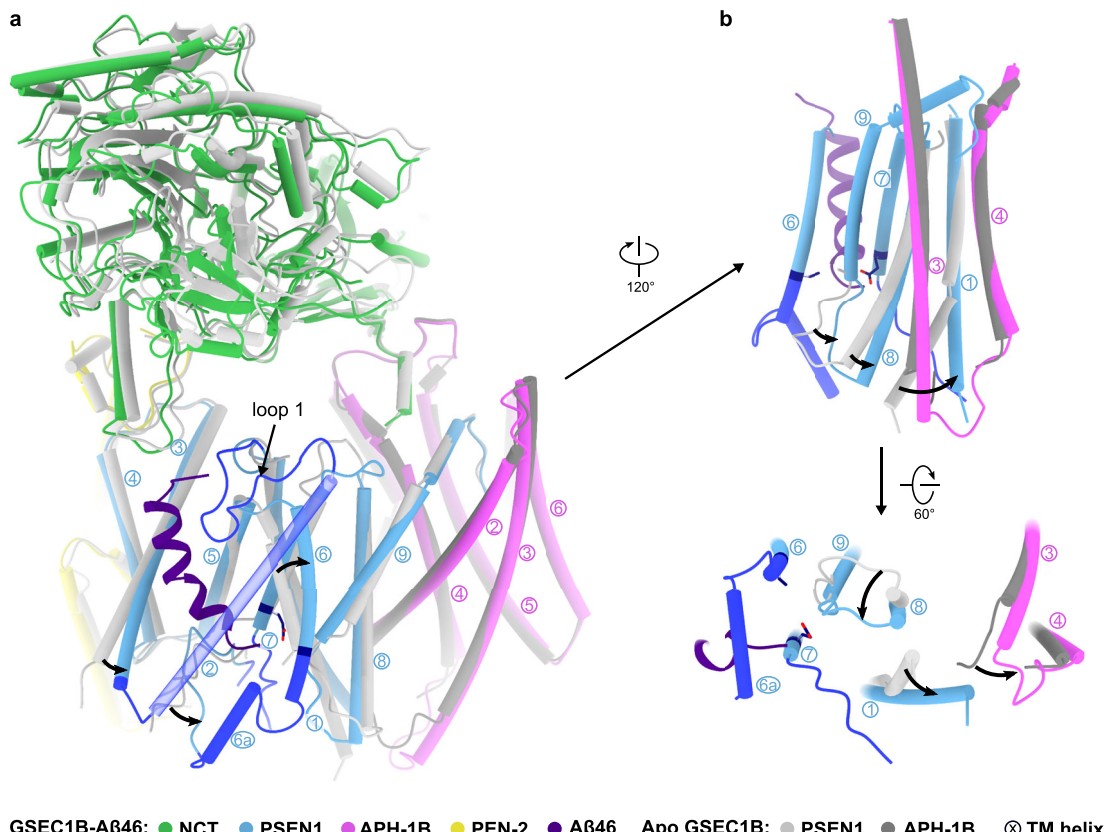

GSEC1B-Aβ46: ● NCT ● PSEN1 ● APH-1B ● PEN-2 ● Aβ46   Apo GSEC1B: ● PSEN1 ● APH-1B ⊗ TM helix

**Fig. 4 | Conformational changes between apo and Aβ46-bound GSEC1B.**
**a** Structural alignment of apo (grey) and Aβ46-bound (colour-coded as in Fig. 2) GSEC1B. PSEN1 and APH-1 TMs are indicated with circled numbers. **b** Structural rearrangement in PSEN1 and APH-1 subunits upon Aβ46 binding. PSEN1 from the apo structure is depicted in light grey, and APH-1 is depicted in dark grey.

include structural signature features, such as residues with long and branched side chains as well as a tandem of glycines (Fig. 6c). We interpret the lack of clear side-chain densities as the superposition of Aβ46 peptides bound with different registers (positions within the binding channel).

**Structural and functional determinants of GSEC-Aβ46 interaction**

The comparison of the GSEC1A-APP$_{C83}$[20] with GSEC1B-Aβ46 complexes showed that the backbones of the substrates closely overlap in the transmembrane region (RMSD of 1 Å over 85 atoms, Fig. 5a–d), but deviate at the extracellular and cytoplasmic interfaces.

On the extracellular side, and consistent with other substrate-bound structures[20,21], an additional density consistent with the bound N-terminus of Aβ46 is observed near Glu650 on the surface of the NCT ectodomain (Fig. 2c). Furthermore, the backbone of the extracellular juxtamembrane region of Aβ46 forms a short, extended strand that bends over loop 1 (Fig. 5a, b). This feature contrasts with the straight helical conformation of the backbones of APP$_{C83}$[20] and Notch[21] in this region. When compared with APP$_{C83}$, the backbone of the first common α-helical turn of Aβ46 bends by about 2.2 Å toward loop 1 (Fig. 5a). This relative shift appears to be the result of the pronounced conformational differences in loop 1, relative to the cross-linked GSEC-APP$_{C83}$/Notch structures which were obtained using a Q112C PSEN1 and Notch/APP$_{C83}$ P1728C/V695C mutants (Fig. 5a–b, d and Supplementary Movie 3).

Loop 1 regulates GSEC proteolysis, mediates the binding of allosteric GSEC modulators, and harbours 23 FAD-linked pathogenic mutations (https://www.alzforum.org/mutations/psen-1, accessed on 28 June 2023). In the GSEC1B-Aβ46 complex, the tip of loop 1 (Tyr115) is inserted 4.5 Å deeper into the substrate-binding transmembrane channel and points towards Aβ46 (Fig. 5a, b, d), such that the hydroxyl group of Tyr115$^{PSEN1}$ is positioned ~ 4.7 Å from the substrate backbone and reaches approximately the middle of the bilayer (Fig. 5b, d). To investigate a possible interaction, we mutated Tyr115$^{PSEN1}$ to Phe and Ala, and rescued (WT or mutant) PSEN1 expression in PSEN1/PSEN2 deficient cells. The presence of mature, glycosylated NCT and N- / C-terminal fragments of PSEN1 demonstrated the efficient reconstitution of GSEC complexes in all generated cell lines (Fig. 5e). To determine the effects of these substitutions in PSEN1 on APP processing, we transiently expressed the APP$_{C99}$ substrate in the WT/mutant cell lines and measured secreted Aβ 37/38/40/42/43 peptides in the conditioned media by ELISA. Removal of either the hydroxyl group (Tyr to Phe mutation) or the aromatic ring of Tyr115 (Tyr to Ala mutation) shifted the short-to-long Aβ (37+40+38)/(43+42) peptide ratio (Fig. 5f). Because this ratio provides an estimation of GSEC processivity[35], the decrease relative to the WT points towards the involvement of Tyr115$^{PSEN1}$ in hydrogen bonding and van der Waals interactions that stabilise the GSEC-Aβ complex during its sequential cleavage.

In the extracellular leaflet, the TM domain of Aβ46 adopts an α-helical conformation, with three helical turns partially exposed to lipids and partially to PSEN1. Starting from the 4th turn, similar to the structures of APP$_{C83}$/Notch[20,21] Aβ46 is enclosed by PSEN1, its helical pitch elongates and transitions to an extended strand conformation (Fig. 5d, Fig. 6a). Notably, the disrupted intra-helical hydrogen bonds within the Aβ46/APP$_{C83}$/Notch substrates are compensated by interactions with hydrogen-donor side chains in PSEN1: Ser169 and Trp165 (Fig. 5d). The contribution of these interactions to the stability of enzyme-substrate interactions, and therefore GSEC processivity[18], remains unknown.

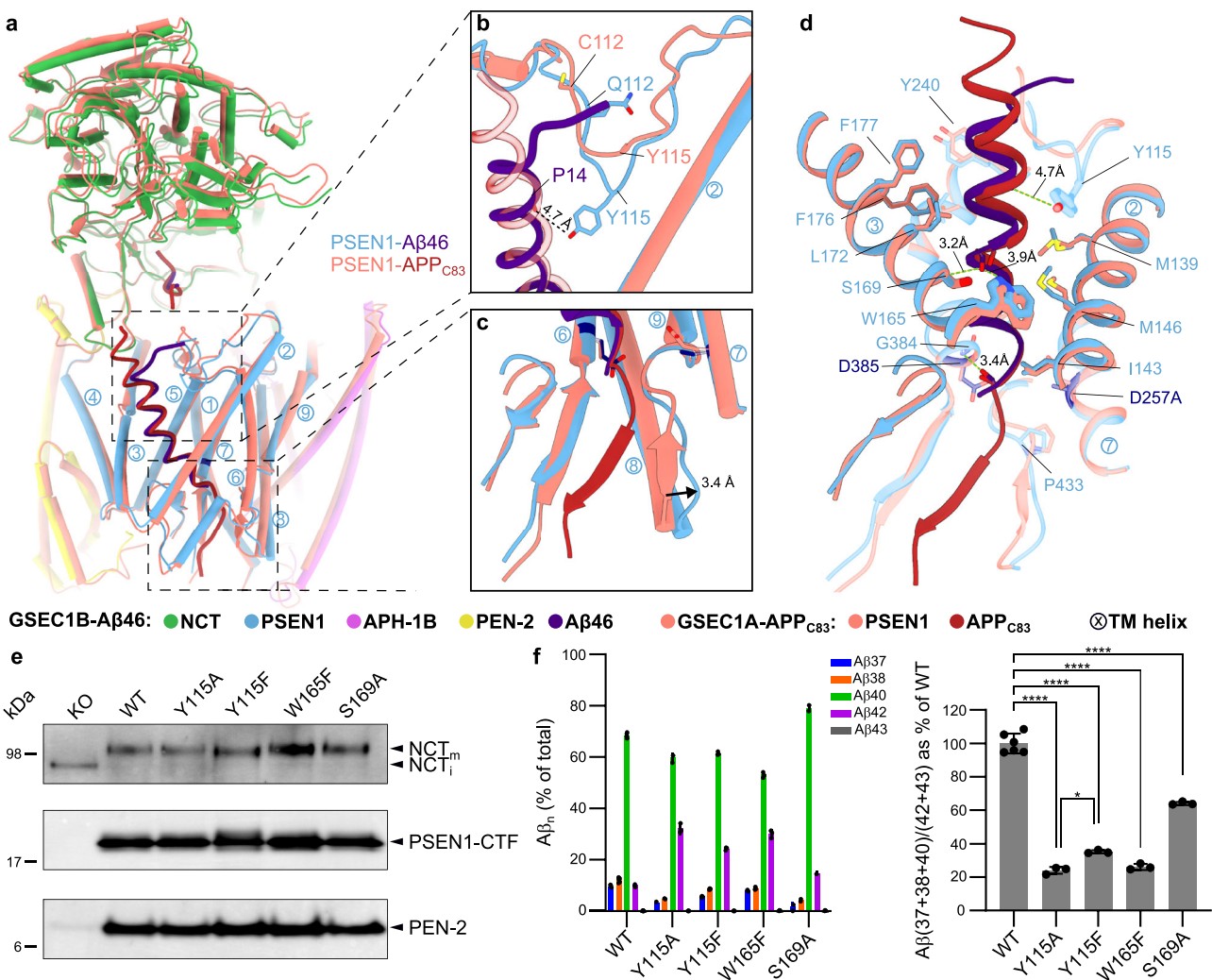

**GSEC1B-Aβ46:** ● NCT ● PSEN1 ● APH-1B ● PEN-2 ● Aβ46 **GSEC1A-APP_C83:** ● PSEN1 ● APP_C83 ⊗TM helix

**Fig. 5 | Structural comparison of GSEC1B-Aβ46 with GSEC1A-APP_C83 and experimental validation of potential hydrogen bonds between PSEN1 and APP.** **a** Structural alignment of GSEC1B-Aβ46 and GSEC1A-APP_C83 (PDB: 6IYC; shown in grey) complexes. PSEN1 TMs are indicated with circled numbers. **b** Closeup of extracellular side of the substrate and loop 1. The GSEC1A-APP_C83 complex was stabilised by disulphide cross-link between V7C APP_C83 (unresolved) and Q112C PSEN1. **c** Closeup view on intracellular side of substrate binding site. **d** Details of PSEN1-Aβ46 interactions in the trans-membrane region. Potential hydrogen bond interactions between the substrates and W165, S169 and G384 are indicated.

**e** Western blot analysis of solubilised membranes from $Psen1^{-/-}/Psen2^{-/-}$ (dKO) mouse embryonic fibroblast cell lines rescued with WT or mutant PSEN1. $NCT_m$ and $NCT_i$ indicate mature glycosylated and immature NCT, respectively. Molecular weights of protein standards are indicated on the left. **f** GSEC processivity of APP_C99 in $Psen1^{-/-}Psen2^{-/-}$ MEFs rescued with WT or mutated PSEN1. Data are presented as mean ± SD, $n = 6$ for the WT and $n = 3$ for the mutants. Multiple comparison ANOVA was used to determine statistical significance ($P < 0.05$); P(WT vs Y115A) < 0.0001, P(WT vs Y115F) < 0.0001, P(WT vs W165F) < 0.0001, P(WT vs S169A) = 0.0001, $P$(Y115A vs Y115F) = 0.0115. Source data are provided as a Source Data file.

To assess their functional relevance, we generated W165F and S169A PSEN1 mutants that are unable to form hydrogen bonds and stably expressed them in PSEN1/PSEN2 deficient cells. Western blot analysis of NCT maturation and PSEN1 endoproteolysis showed that both mutants efficiently reconstituted mature GSEC complexes (Fig. 5e), while the ELISA-based analysis of secreted Aβ profiles demonstrated that these mutations impaired GSEC processivity of APP_C99. These results support the involvement of both residues in Aβ substrate stabilisation and, therefore, GSEC processivity (Fig. 5f).

The GSEC-Aβ46 structure also showed that a hydrogen bond between a carbonyl oxygen in the C-terminally extended backbone of the substrate and the backbone nitrogen of Gly384[PSEN1] is preserved in both initial and intermediate substrates (Fig. 5d). The latter is part of the conserved active site GX**G**D motif of aspartyl intramembrane proteases[36] (Fig. 5d). The AD-linked G384A[PSEN1] mutation[37,38] destabilises GSEC-Aβ interactions[18,39], supporting a stabilising role of the hydrogen-bonding interaction between Gly384[PSEN1] and the substrate's backbone.

On the cytoplasmic side, the density of Aβ46 ends with a two-residue stretch in an unwound conformation next to the catalytic Asp385[PSEN1] (Figs. 2 and 5). In contrast, the initial endopeptidase substrates[20,21] have a 6-residue-long ordered extension in a β-strand conformation (Fig. 6b) that forms the hybrid β-sheet, contributed by PSEN1 TM6-TM7 loop (residues 287–289 and 379–381) on one side and TM8-TM9 loop (residues 430–432) on the other (Fig. 5c). In the GSEC1B-Aβ46 model, the PSEN1 TM8-TM9 loop is shifted away by 3.4 Å from the remaining two β-strands (Fig. 5c).

Collectively, these data show that the substrate conformation is largely similar between initial and intermediate cuts, suggesting that GSEC shapes the substrate as the substrate rearranges during processive proteolysis at its C-terminus. The structural data also reveal important rearrangements in the E-S complex that involve the loop 1[PSEN1] and the extracellular end of the Aβ46 TM. In addition, the E-S bonding is stabilised by several polar interactions which likely also facilitate the unwinding of the substrate α-helix in the membrane.

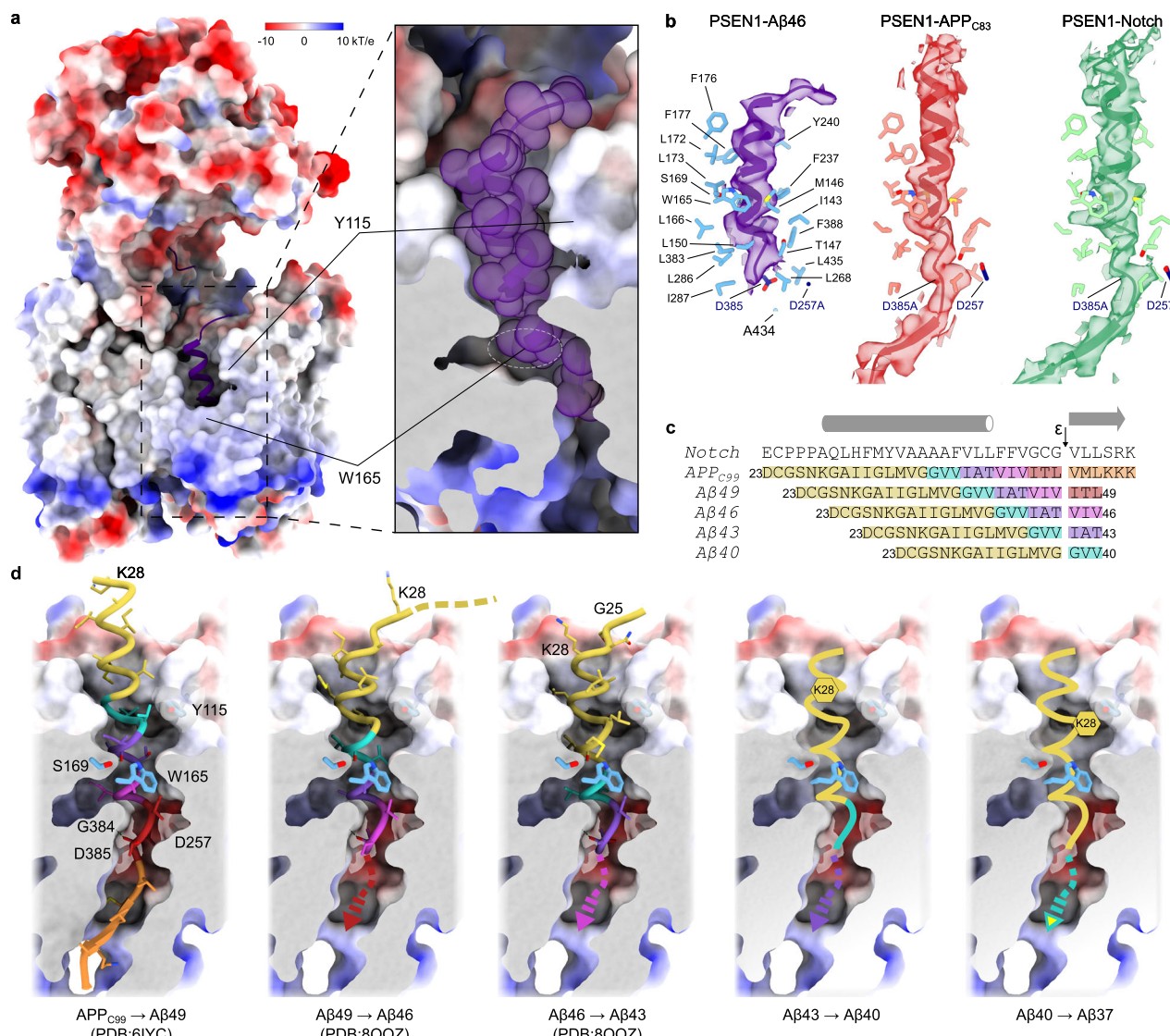

**Fig. 6 | Substrate interaction with GSEC and the model of sequential catalysis.**
**a** Surface representation of GSEC coloured by electrostatic potential and
Aβ46 shown as cartoon. Fenestration in the intracellular membrane leaflet region of
PSEN1 partially exposes Aβ46 to the membrane environment. **b** Structure of
substrate-binding channel of PSEN1 is identical with three different substrates and
the conformations of the three substrates are very similar. **c** Sequence alignment of
Notch and the APP$_{C99}$ downstream products along the Aβ49 pathway. The initial
endopeptidase cleavage site is indicated with the arrow and different colours
indicate the tripeptides sequentially cleaved in the Aβ40 product line. **d** A model of
sequential catalysis. The structures of substrates in positions corresponding to the
cuts producing known Aβ peptides are shown. Panels 1 and 3 counted from the left
side are experimental structures, the remaining panels are models.

## Discussion

Close to 300 mutations in GSEC cause early-onset familial AD. These
pathogenic mutations destabilise the interaction between GSEC and
APP, thus enhancing the release of longer and more hydrophobic Aβ
peptides. Molecular understanding of the mechanisms underlying the
modulation of the Aβ length will facilitate the development of drugs
safely and efficiently targeting GSEC in AD therapy. To gain insights
into the mechanisms modulating and defining Aβ length, we deter-
mined high-resolution structures of GSEC complexes containing the
APH-1B isoform in apo state and bound to Aβ46, then we interrogated
GSEC-Aβ interactions to identify the key determinants of E-S stability.

APH-1 serves as an essential scaffold during the assembly of the
GSEC complex[40,41] and modulates its proteolytic activity in an isoform-
dependent manner[11]; APH-1A type GSECs generate shorter Aβ pro-
ducts, relative to the APH-1B type GSECs. Comparison of GSEC1B and
the reported GSEC1A[20,28] structures revealed that differences in APH-1
are confined to regions with low sequence conservation between the
isoforms (Fig. 3e). These differences are consistent between the apo

(Fig. 3) and substrate-bound GSEC structures (Supplementary Fig. 8a),
regardless of the used lipid-mimetic environments (lipid nanodiscs,
amphipols and digitonin). These findings suggest that the structural
differences in APH-1 are sequence- rather than environment-depen-
dent, which is further corroborated by the APH-1 structures predicted
by AlphaFold2 (Supplementary Fig. 8b).

Structurally variable regions in APH-1 isoforms (Fig. 3e) map to the
interface with PSEN1. Particularly interesting is the TM3-TM4$^{APH-1}$ loop
which changes conformation upon substrate binding and moves
concertedly with cytosolic ends of TM1$^{PSEN1}$ and TM8$^{PSEN1}$ (Fig. 4b).
These TMs in PSEN1 play a key role in substrate cleavage[30,32,33]. Even
though the conformation of TM1$^{PSEN1}$ is identical between substrate-
bound GSEC1A and GSEC1B, the APH-1 isoforms may alter the
dynamics of this structural triad (TM3-TM4$^{APH-1}$ loop, TM1$^{PSEN1}$ and
TM8$^{PSEN1}$), thereby changing enzyme processivity. As noted, the AD-
linked P88L mutation in TM1$^{PSEN1}$ [34], which replaces the helix-breaking
Pro with a helix-forming Leu, promotes the release of (very) long Aβ45/
46 peptides[33]; thus, suggesting that dynamic hinge-like motions of

TM1[PSEN1] are necessary for efficient sequential catalysis. Since the charge and length of the APH-1 TM3-TM4 loop differ between isoforms, this loop might exert allosteric-like effects by directly modulating the dynamics of TM1 [PSEN1] or indirectly via TM8[PSEN1]. Although these structural differences are intriguing, it is unclear whether they explain the differential activities associated with the APH-1 A/B isoforms. Functional analysis is needed to clarify this.

Remarkably, the structure of the substrate-binding transmembrane channel within PSEN1, that includes the conformation of side chains of the TM residues lining the channel, is virtually identical in the different GSEC-substrate complexes with Aβ46, APP[C83][20] and Notch[21] (Fig. 6b). This indicates that the overall substrate recognition, and associated conformational changes in PSEN1 and substrate, are largely independent of the substrate sequence (Fig. 6c) and length and thus are likely defined by GSEC.

However, the conformation of the functionally important loop 1 is different in the GSEC-Aβ46 complex, relative to the APP[C83] and Notch structures[20,21] (Fig. 5b and Supplementary Fig. 9). The differences are likely caused by the engineered cysteine bridge used to stabilise APP[C83] and Notch substrates within GSEC1A (Supplementary Fig. 9a). This is supported by the similar conformations of loop 1 in the inhibitor-bound[26–28] and GSEC-Aβ46 structures (Supplementary Fig. 9b).

Similar to APP[C83] and Notch, the Aβ46 helix is partially unwound close to the scissile bond and stabilised by plausible hydrogen bonds between its backbone and PSEN1 sidechains Trp165, Ser169 and Gly384 (Fig. 5a, d). Additionally, in the unperturbed conformation of loop 1, Tyr115 points in the direction of substrate suggesting an E-S polar interaction inside the membrane bilayer (Fig. 5b). Because we have previously shown that GSEC-Aβ complex stability defines enzyme processivity and Aβ length[18], we investigated the contribution of the polar Tyr115, Trp165 and Ser169 residues to the processing of APP[C99] in cells (Fig. 5e, f). Removal of polar groups by mutagenesis reduced processivity in all instances supporting a significant stabilising effect of the E-S hydrogen bonds. Furthermore, for Tyr115 even stronger reduction of processivity was observed after removal of the aromatic ring pointing towards contribution of Van der Waals interactions to the stabilisation of E-S interactions.

These data explain the pathogenic increases in longer Aβs by FAD mutations targeting these positions (Y115H, Y115C, W165C, S169P, S169L, ΔS169[42–50]). Moreover, our previous studies have shown that the FAD-linked G384A mutation generates longer Aβ peptides[18]. All of these mutations eliminate the hydrogen-bond-donor sidechains or change the location of the hydrogen-bond-donor atoms, thus suggesting that they reduce processivity by disturbing the hydrogen bonds stabilising the partially unwound Aβ α-helix.

In the structures of GSEC1A-APP[C99] or -Notch, the substrates downstream from the scissile bond had extended conformations and contributed a strand to a hybrid β-sheet[20,21] (Fig. 5c). The latter is proposed to play important roles in stabilising substrate binding and orienting the scissile peptide bond towards the catalytic aspartates[21]. In GSEC1B-Aβ46 structure, the density of the substrate stops after the scissile bond, and the density of the hybrid β-strand – previously seen for endopeptidase substrates – is absent. This raises the following scenarios: the carboxypeptidase-like cleavages do not require the formation of the hybrid β-sheet, or the hybrid β-sheet is absent because the equilibrium of distribution of bound Aβ46 in the substrate-binding transmembrane channel is shifted towards Aβ46 bound as a product generated upon Aβ49 cleavage, rather than as a substrate for the Aβ46 → Aβ43 cut (Fig. 6a, d). We believe that the first scenario is unlikely given that a formation of a β-sheet between substrate and protease is found in almost all studied proteases[51] and given a pattern of 3 to 4 amino acids cut steps.

It remains unclear how substrate rearranges its structures to deliver amino acids located 9 to 12 positions upstream from the first cut to the catalytic site (Fig. 6c). The following models are suggested: (1) a piston model in which the transmembrane helical region of the substrate shifts into the hydrophobic region of PSEN1 channel, bringing polar substrate region inside the membrane; (2) a model according to which the transmembrane helical region gradually unwinds[21] or (3) both, progressive C-terminal unwinding of the transmembrane helical region with catalysis and partial substrate threading (piston movement)[18].

In GSEC1B-Aβ46 structure, the extracellular part of the transmembrane helix of Aβ46 is embedded by ~ 4.5 Å (nearly one α-helical turn) deeper into the channel, relative to the initial APP[C83]/Notch substrates. Moreover, it is tilted towards loop 1 to position an unstructured juxtamembrane segment of Aβ46 over loop 1[PSEN1].

Our structure supports the piston model, at least during the first steps of the processive proteolysis, schematically shown in Fig. 6d. As a consequence of the Aβ α-helix embedding into the substrate channel, positively charged Lys28 must move inside the hydrophobic substrate channel for the Aβ40 → Aβ37 cut (Fig. 6d). Even though this process might be facilitated by the loop 1 and Tyr115, embedding a charged residue into a hydrophobic environment is energetically costly. Consistently, the fraction of released Aβ37 is low as compared to Aβ40[52]. This mechanism is also supported by reports in literature showing that the K28A mutation causes release of very short Aβ species[53,54], as well as our recent data demonstrating that charged/polar residues in the ectodomain of the APP limit substrate threading and promote product release[55].

In conclusion, our structural and functional analyses provide structural insights into the allosteric-like modulation of GSEC by the APH-1 isoforms, outline the interaction of Aβ46 with GSEC and define the contribution of polar E-S interactions for efficient processive catalysis, including the additionally identified interaction between the substrate and PSEN1 loop 1.

Regarding the allosteric-like role of APH-1, our structural data define 'regions of interest' in APH-1 for the design and functional analysis of APH-1A/B chimeras that test the potential involvement of these regions in the allosteric-like modulation by the APH-1 isoforms. Further, our GSEC-Aβ46 structure represents a first step towards structural understanding of GSEC sequential catalysis. To fully understand the sequential mechanism, the elucidation of the (γ) E-S co-structures, involved in the different stages of the GSEC-mediated APP cleavage is required, as this will provide key insights into the recognition and stabilisation of Aβ peptides during sequential proteolysis. The challenge of achieving these structures is, however, significant, given the increasingly higher aggregation propensities of Aβ peptides longer than Aβ42, and the decreasing affinity of peptides shorter than Aβ46[18]. Therefore, structures and functional data, together with the methods described here will facilitate fundamental research into the mechanisms underlying GSEC-mediated proteolysis and may deliver frameworks for the discovery of drugs that safely and efficiently tackle toxic Aβ production.

## Methods

### GSEC1B expression, purification and formation of GSEC1B-Aβ46 complex

Human NCT, PSEN1, APH-1B and PEN-2 were expressed in High Five insect cells using a baculovirus expression system as previously described[11]. NCT was cloned with a PreScission protease cleaving site and GFP tag at the C-terminus. The same system was used to express inactive GSEC1B complex (GSEC1B[D257A]) in which PSEN1 was expressed as N- and C-terminal fragments (amino acids 1-297 and 298-467, respectively) to mimic the autoproteolytic activation of GSEC.

All the purification steps were carried out at 4 °C. Seventy two hours after infection, cells were collected by centrifugation (4800 x $g$, 20 min) and resuspended in 100 ml of lysis buffer (25 mM PIPES pH 7.4, 300 mM NaCl, 10% glycerol, 1x Complete protease inhibitor cocktail

(Roche)) per litre of culture. Resuspended cells were lysed using Emulsiflex C3 homogeniser (Avestin) and total membrane fractions isolated by ultracentrifugation (100,000 x $g$, 1 h). Membrane pellets were washed twice in 50 ml of high-salt wash buffer (25 mM PIPES pH 7.4, 1 M NaCl, 10% glycerol) per litre of culture; pellets were resuspended using a PTFE plunger in a Heidolph overhead stirrer, incubated on a rotator for 30 minutes and pelleted by ultracentrifugation. Washed membranes were resuspended in solubilisation buffer (25 mM PIPES, 300 mM NaCl, 2% CHAPSO (Anatrace), 5% glycerol, 1 x Complete protease inhibitor cocktail). After overnight incubation at 4 °C, the soluble fraction was separated by ultracentrifugation and incubated overnight with agarose resin NHS-coupled to anti-GFP nanobodies[56] (NHS-activated Sepharose 4 FF; Cytiva). Resin was transferred into a gravity column (Bio-Rad) and washed with 20 column volumes (CV) of solubilisation buffer followed by 10 CV of wash buffer (25 mM PIPES, 300 mM NaCl, 1% CHAPSO, 0.1% 1-palmitoyl-2-oleoyl-glycero-3-phosphocholine (POPC; Avanti), 5% glycerol) and 10 CV of elution buffer (25 mM PIPES, 150 mM NaCl, 1% CHAPSO, 0.1% POPC, 1 mM dithiothreitol (DTT), 1 mM ethylenediaminetetraacetic acid (EDTA)). Next, the resin was resuspended in 1 CV of elution buffer and GSEC1B was eluted by overnight incubation with 50 µg/ml of PreScission protease[57]. To remove PreScission protease, the eluted fraction was incubated overnight with Glutathione Sepharose 4B resin (Cytiva). Protein concentration was estimated using Bradford reagent (Bio-Rad) following the manufacturer's instructions and was in the 0.7–1.5 mg/ml range. Protein purity was assessed by SDS-PAGE (4-12% Bis-Tris; Invitrogen) and Coomassie staining (InstantBlue; Abcam). Purified protein was flash-frozen and stored at −80 °C.

To form GSEC1B-Aβ46 complex, 5 µM Aβ46 (rPeptide) resuspended in dimethyl sulfoxide (DMSO) was added to purified GSEC1B$^{D257A}$ (1.25 x fold excess), followed by a 1 h incubation at 37 °C.

### Expression and purification of membrane scaffold protein

Plasmid with MSP1D1, pMSP1D1[58] (Addgene) was transformed into *E. coli* BL21(DE3) competent cells. Culture was grown in LB medium containing 25 µg/ml kanamycin at 37 °C until OD$_{600}$ of 0.8. Protein expression was induced by adding 1 mM IPTG and carried out for 3 h. Cells were collected by centrifugation (4800 x $g$, 20 min, 4 °C) and the pellet stored at −20 °C.

All the purification steps were carried out at 4 °C. Cell pellet was resuspended in 50 ml of lysis buffer (150 mM Tris pH 8, 300 mM NaCl, 20 mM imidazole pH 8, 2.5 mM MgCl$_2$, 0.1 mM CaCl$_2$) per litre of culture and incubated with a few grains of DNAse I (Sigma) for 1 h. Resuspended cells were lysed using a continuous flow cell disruptor (Constant Systems) and supplemented with 1% Triton X-100. Cell lysate was centrifuged (40,000 x $g$, 20 min), and the supernatant was filtered through a 0.45 µm filter before it was applied onto a HisTrap HP (Cytiva) column using an ÄKTA pure system (Cytiva). The column was washed with 10 CV of wash buffer 1 (40 mM Tris pH 8, 300 mM NaCl, 20 mM imidazole pH 8, 1% Triton X-100), 10 CV of wash buffer 2 (40 mM Tris pH 8, 300 mM NaCl, 20 mM imidazole pH 8, 2.15% sodium cholate (Sigma), 1% Triton X-100), 10 CV of wash buffer 3 (40 mM Tris pH 8, 300 mM NaCl, 20 mM imidazole pH 8, 1% sodium cholate), and 10 CV of wash buffer 4 (40 mM Tris pH 8, 300 mM NaCl, 50 mM imidazole pH 8, 1% sodium cholate). MSP1D1 was eluted using an imidazole gradient (from 50 to 400 mM) over 10 CV in wash buffer 4. The peak fractions containing MSP1D1 were pooled and dialysed overnight in 3.5 kDa MWCO cellulose tubing (Carl Roth) against 80 x volume of dialysis buffer (40 mM Tris pH 8, 150 mM NaCl) to remove excess of imidazole. Protein concentration was measured spectrophotometrically ($\varepsilon_{280}$ 21430 M$^{-1}$cm$^{-1}$, MW 24.79 kDa) using NanoDrop One (Thermo Scientific). Dialysed MSP1D1 was concentrated to 5 mg/ml using a 10 kDa MWCO Amicon Ultra centrifugal concentrator (Millipore), supplemented with CHAPSO (4%) and DTT (2 mM) and, to cleave the His-tag, the sample was incubated overnight with TEV

protease (1 µg of TEV protease per 100 µg of MSP). The TEV protease was expressed and purified as previously described[59]. Cleaved MSP1D1 was applied onto a HisTrap HP column (Cytiva), the flowthrough containing untagged MSP1D1 was collected, concentrated to 5 mg/ml, and further purified by size exclusion chromatography using a Superdex 75 column (Cytiva) in SEC buffer (25 mM PIPES pH 7.4, 150 mM NaCl, 0.5% CHAPSO). Protein concentration was measured spectrophotometrically ($\varepsilon_{280}$ 18450 M$^{-1}$cm$^{-1}$, MW 22.04 kDa). Fractions containing pure untagged MSP1D1 were pooled and concentrated to ~10 mg/ml, then flash frozen and stored at -80 °C.

### Reconstitution of GSEC1B in lipid nanodiscs

We screened reconstitution conditions including two constructs of membrane scaffold protein (MSP) and two lipid compositions. The optimal reconstitution was obtained using MSP1D1 and POPC:DLPC (1,2-dilauroyl-sn-glycero-3-phosphocholine; Avanti) mix at 1:1 molar ratio.

All the steps were carried out at 4 °C. Purified GSEC1B or GSEC1B-Aβ46 complex at a concentration of 0.7–1.5 mg/ml was diluted with a solution containing 10% CHAPSO and 1.67% DLPC to a final concentration of 1.2 mM POPC and 2.4 mM DLPC and the solution was stirred for 30 min. MSP1D1 was added to the solution to obtain the MSP:POPC:DLPC molar ratio of 1:60:80 and the solution was stirred for 1 h. Reconstitution of nanodiscs was achieved upon detergent adsorption by Bio-Beads SM-2 resin (Bio-Rad) which was added in three batches, 0.25 g/ml each, and the protein solution was incubated while stirring for 1 h after the first batch, overnight after the second, and 1 h after the third. Nanodiscs were collected from the tube by centrifugation after puncturing the bottom of the tube. The reconstituted GSEC1B in nanodiscs was separated from empty nanodiscs by size exclusion chromatography using a Superdex 200 Increase column (Cytiva) in SEC buffer (10 mM PIPES pH 7.4, 100 mM NaCl). SDS-PAGE and silver staining (Pierce silver stain kit; Thermo Scientific) were used to assess the contents of the fractions.

Peak fractions containing GSEC1B were concentrated to 0.04–0.1 mg/ml using a 100 kDa MWCO Amicon Ultra centrifuge filter (Millipore).

GSEC1B and GSEC1B$^{D257A}$ purified in CHAPSO and reconstituted into nanodiscs were resolved by SDS-PAGE in a 4–12% Bis-Tris NuPAGE gel (ThermoScientific) and transferred to a nitrocellulose membrane. The following primary antibodies and dilutions were used for immunoblotting: anti-NCT (BD Biosciences Cat# 612290, RRID:AB_399607) 1 in 2000, anti-PSEN1$_{NTF}$ (Millipore Cat# MAB1563, RRID:AB_11215630) 1 in 1000, anti-PSEN1$_{CTF}$ (Cell Signalling Technology Cat# 5643, RRID:AB_10706356) 1 in 1000, anti-APH-1B (B78; kindly provided by prof. Bart de Strooper) 1 in 1000, and anti-PEN-2 (Cell Signalling Technology Cat# 8598, RRID:AB_11127393) 1 in 500. The following secondary antibodies were used: goat anti-mouse IgG-HRP conjugate (Bio-Rad Cat# 1721011, RRID:AB_2617113) 1 in 10000, rabbit anti-rat IgG-HRP conjugate (Thermo Fisher Scientific Cat# 61-9520, RRID:AB_2533945) 1 in 2000, and goat anti-rabbit IgG-HRP conjugate (Bio-Rad Cat# 172-1019, RRID:AB_11125143) 1 in 10000. Blots were developed using Western Lightning Plus-ECL Enhanced Chemiluminescence Substrate (Perkin Elmer).

### GSEC1B activity assays in nanodiscs

The activity assays were carried out at 37 °C in a Labcycler Gradient thermocycler (SensoQuest). Final buffer composition of the reactions was 25 mM PIPES, 150 mM NaCl, 0.025% DMSO. GSEC1A was expressed, purified, and reconstituted into nanodiscs as described above. Assays were carried out with 0.23 µM GSEC1A/GSEC1B/GSEC1B$^{D257A}$ and 2.5 µM Aβ46 for 24 h. Reactions were quenched by placing the assay tubes on ice and adding 10 µM inhibitor L-685,458 (Santa Cruz Biotechnology). De novo Aβ (37, 38, 40, 42) production was quantified by MSD ELISA as described previously[52], and Aβ43 was quantified using

Amyloid-beta (1-43) (FL) ELISA kit (IBL) according to the manufacturer's instructions. The plots were generated using GraphPad Prism version 8.4.2.

### Generation of MEF cell lines and the cell-based activity assays

Single-point mutations (Y115A, Y115F, W165F and S169A) were introduced into human PSEN1 cDNA and cloned into the pMSCVpuro vector using Q5 Site-Directed Mutagenesis Kit (NEB BioLabs) according to the standard protocol. The sequences of the primers used are available in Supplementary Data 1. To generate recombinant retroviruses, the generated vectors and the PIK packaging plasmid were delivered into HEK 293 T cells using the TransIT-LT1 transfection reagent (Mirus Bio). Stable MEF cell lines expressing either WT or mutant PSEN1/GSEC complexes were generated through retroviral transduction of *Psen1$^{-/-}$/Psen2$^{-/-}$* mouse embryonic fibroblasts (MEFs)[60] as previously described[52]. MEFs were cultured in Dulbecco's Modified Eagle's Medium (DMEM)/F-12 (Life Technologies) supplemented with 10% foetal bovine serum (FBS) (Sigma-Aldrich)[35]. Cell lines stably expressing the WT/mutant proteins were selected using media supplemented with 5 µg/ml puromycin (Sigma-Aldrich). The reconstitution of mutant GSECs was assessed by SDS-PAGE and western blot. Briefly, cells were collected, and total membranes isolated and solubilised in 28 mM PIPES pH 7.4, 210 mM NaCl, 280 mM sucrose, 1.5 mM EGTA pH 8, 1% CHAPSO, 1x Complete protease inhibitor cocktail. Equal amounts of solubilised protein were resolved in a 4–12% Bis-Tris NuPAGE gel (ThermoScientific) and transferred to a nitrocellulose membrane. The following primary antibodies were used: anti-NCT (9C3; kindly provided by Prof. Wim Annaert) 1 in 2000, anti-PSEN1$_{CTF}$ (as described above), and anti-PEN2 (as described above). Secondary antibodies used and blot development was done as described above.

To assess the effects of PSEN1 mutants on Aβ production, APP$_{C99}$ substrate was transiently expressed in the generated MEF cell lines using a recombinant adenoviral expression system as previously described[61]. In brief, cells were plated in a 96-well plate at the density of 12,500 cells/well and transduced 6-8 h later with Ad5/CMV-APP$_{C99}$ adenovirus[52]. The culture medium was replaced with a low-serum medium (DMEM/F-12 medium containing 0.2% FBS) 16 h later, and collected after a 24 h incubation period at 37 °C.

The conditioned media was cleared by centrifugation at 800 x *g* for 15 min and used to determine Aβ (37, 38, 40, 42) peptide levels using MSD ELISA as described previously[52], and Aβ43 was quantified using Amyloid-beta (1-43) (FL) ELISA kit (IBL) according to the manufacturer's instructions. The plots were generated using GraphPad Prism version 8.4.2.

### Preparation of graphene oxide coated EM grids

Carbon-coated holey grids Quantifoil R 0.6/1 Cu 300 and CF-2/1-3 C (EMS) were used for preparing cryo-EM samples for apo GSEC1B and GSEC1B-Aβ46 complex, respectively. Graphene oxide 0.2% (w/v; Sigma) and 1% (w/v; GOgraphene) were used interchangeably.

Grids were glow-discharged with carbon face up using ELMO glow discharge system (Agar Scientific) with 5 mA current for 1 min at 0.3 mbar in air. A volume of 4 µl of 0.5 mg/ml poly-L-lysine solution (MW 15–30 kDa; Sigma) in 10 mM PIPES pH 7.4, 100 mM NaCl was pipetted on the carbon side of the grid and incubated for 2 min, blotted with Whatman grade 2 paper and washed twice by pipetting 4 µl of MilliQ water on the carbon side, followed by blotting. After 5 min drying on air, 3 µl of 0.2 mg/ml graphene oxide was pipetted on carbon side of the grid and incubated for 2 min. Next, the grid was blotted, and washed three times by touching a droplet of water with the GO side, followed by blotting. The grid was dried for 30 min and 4 µl of 0.1% PEG 10,000 (Fluka Chemica) in 10 mM PIPES pH 7.4 and 100 mM NaCl was applied on the carbon side of the grid and incubated for 2 min. The grid was blotted and washed twice by pipetting 4 µl of water on top and blotting. The grid was dried for 5 min and used for preparation of cryo-EM samples immediately.

### Cryo-EM sample preparation

GSEC1B nanodisc solution (2 µl) at concentration of 0.04 mg/ml was pipetted on the front side of graphene-oxide-coated ('MRC protocol'[62]) Quantifoil R 0.6/1 Cu300 grid and incubated for 15-30 sec at 99% humidity in Cryoplunge 3 (Gatan). The grid was blotted from both sides for 2.5-3 sec with Whatman grade 3 paper and plunged into liquid ethane at -175 °C.

For the GSEC1B-Aβ46 complex, 5 µM Aβ46 (15-fold excess) resuspended in DMSO was added to GSEC1B-Aβ46 reconstituted into lipid nanodiscs (1% final DMSO concentration) and the mix was incubated for 1 h at 37 °C. Next, the protein solution was cooled on ice and plunge frozen on CF-2/1-3 C grids coated with graphene oxide (according to the protocol described in the section above) as described for the apo protein complex.

### Cryo-EM data collection

The micrographs were collected on a CRYO ARM 300 electron cryogenic microscope (JEOL) equipped with an in-column Omega energy filter and operated at 300 keV. The energy filter slit was set to 20 eV and the data were collected at a nominal magnification of 60,000 with a magnified pixel size of 0.766 Å on a K3 direct electron detector (Gatan). SerialEM (3.8.2 for apo, 3.8.18 for Aβ46-bound data) was used for automated data collection.

The micrographs for GSEC1B in apo state were collected as 60-frame movies with an electron dose of 1.06 e$^-$ Å$^{-2}$ per frame over 3 seconds in a 5 × 5 pattern with one exposure per 0.6 µm nominal and 0.3 µm measured hole diameter, producing 25 micrographs per stage position. A total of 10,733 movies were collected from one EM grid with the defocus in the range from −1.2 to −2 µm.

For the GSEC1B-Aβ46 complex the micrographs were collected in a 3 × 3 pattern with 5 exposures per each 2 µm hole resulting in 45 movies per stage position. Each movie contained 60 frames with total exposure of 2.8 s and an electron dose of 0.936 e- Å$^{-2}$ per frame. A total of 18,855 movies were collected. The defocus was set to −1.3 to −2.3 µm.

### Image processing

Movies were pre-processed on-the-fly using RELION 3.1 schedules[63] as a wrapper to run MotionCor2 1.4.2[64] for frame alignment and CTF parameters were estimated using CTFFIND 4.1.14[65]. Further processing was done using RELION 3.1 unless otherwise stated. Particles from a subset of micrographs were auto-picked using crYOLO 1.7[66] with the general model and submitted to 2D classification in either RELION or cryoSPARC (v3.2.0 for the apo dataset, v3.3.1 for the Aβ46-bound dataset)[67]. 100 micrographs containing the highest number of particles from good classes were manually screened to remove bad particles, then these micrographs and particle coordinates were used to refine the crYOLO general picking model. The refined model was used to auto-pick the entire dataset.

For the apo dataset, 986,830 particles were picked from 5501 micrographs and extracted in a 320 × 320 pixel box downsampled to 64 × 64 pixels (Supplementary Fig. 3). 2D classification was done in RELION (ignoring CTFs until the first peak) and in cryoSPARC (setting initial classification uncertainty factor to 20, online-EM iterations to 100, final full iterations to 20, batch size per class to 1000, enforcing non-negativity, activating clamp-solvent option, disabling FRC-based regularizer, full FRC, setting iteration to start annealing sigma to 10, number of iterations to anneal sigma to 50, and using white noise model). Particles belonging to good 2D classes from both programs were merged and duplicate particles removed resulting in a total of 790,832 particles which were then reextracted in a 320 × 320 pixel box downscaled to 128 × 128 pixels. Ab-initio reconstruction in cryoSPARC

was used to generate an initial model from a subset of particles. 3D classification with 1 class and using tau_fudge value of 64 was used to centre and align the particles. 3D classification with 4 classes and tau_fudge value of 64 resulted in 2 classes with visible transmembrane helices accounting for 413,321 particles which were then reextracted in a 320 × 320 pixel box downscaled to 256 × 256 pixels. 3D auto-refinement of these particles resulted in a map resolved to an average resolution of 3.8 Å. Next, 3D classification without alignment with 2 classes and tau_fudge value of 8 was used to separate particles contributing to high-resolution reconstruction and resulted in 115,197 particles. A 3.6 Å map was reconstructed from these particles which was improved to 3.3 Å after reextracing in a 320 × 320 pixel box without downsampling, Bayesian polishing and defocus refinement. To better resolve the transmembrane region, another round of 3D refinement was performed using external reconstruction with SIDESPLITTER[68] which resulted in the final map at a resolution of 3.2 Å. The final map was sharpened using post process procedure in RELION. The pixel size was calibrated using the NCT ectodomain from the GSEC1A-C83 structure (PDB: 6IYC) as a reference in UCSF Chimera[69], by fitting the reference model into the experimental map. The final map was rescaled and sharpened using the calibrated pixel size of 0.776 Å yielding a 3.3 Å map. The map was further filtered using local resolution filter and the B-factor determined during post processing. The local filtered map was used for model building and deposited to EMDB.

For the GSEC1B-Aβ46 complex, 2,788,683 particles were picked from 18,855 micrographs and classified in 2D as described above (Supplementary Fig. 5). 2,433,778 particles selected after 2D classification were centred and aligned using a 3D classification with 1 class and using tau_fudge value of 64. 3D classification with 10 classes, tau_fudge value 64 over 75 iterations was done using a low-pass filtered map from the apo dataset as a reference model. 3D classes with visible transmembrane helices from iterations 51-75 were picked and duplicate particles removed for a total of 2,023,687 particles which were then subjected to 3D refinement to obtain a map resolved to 4.1 Å. The densities for PSEN1 TM2 and Aβ46 were very weak. Further 3D classification of these particles (K = 10; T = 4; 3.7° sampling; 15° search range; 25 iterations) resulted in two high-resolution classes containing 397,175 particles which were subjected to 3D refinement and resulted in a 3.5 Å map with improved, but still weak densities of PSEN1 TM2 and Aβ46. CTF refinement including anisotropic magnification, beam tilt, trefoil, per particle defocus and per micrograph astigmatism followed by Bayesian polishing improved the resolution of the reconstruction to 3.3 Å. A mask containing PSEN1 and Aβ46 density was used to subtract the rest of the signal and the subtracted particles were subjected to 3D classification without alignment (K = 10; T = 32; 400 iterations) which resulted in a class containing 53,612 particles with improved densities of PSEN1 TM2 and Aβ46. These particles were reverted and subjected to 3D refinement which yielded a map resolved to 3.4 Å. The pixel size was calibrated to 0.949 (320 × 320 pixel box downscaled to 256 pixels) and the map was postprocessed, sharpened and filtered in the same way as described above, yielding a 3.4 Å map. The local filtered map was used for model building and deposited to EMDB.

### Model building and refinement

For the apo structure, the starting model was obtained from the structure of GSEC1A in amphipols (PDB: 5FN5)[28] and rigid-body-fitted into the EM density using UCSF Chimera[69]. A model of APH-1B was built with by homology modelling using modeller 9[70]. The model was further manually built and refined using Coot 0.9.8[71] after which it was refined using real-space refinement with simulated annealing in PHENIX 1.19.2[72]. Several iterations of manual rebuilding in Coot and automated refinement in PHENIX with secondary structure restraints were used to refine the model. The model was validated using MolProbity[73].

For the GSEC1B-Aβ46 structure, the starting model was obtained from the structure of GSEC1A-C83 (PDB: 6IYC)[20] from which the sugars, substrate and lipids were removed. Aβ46 was manually built as a polyalanine model. PSEN1 loop 1 (amino acids 106-123) was manually built to fit into the density and the resulting PSEN1 model was used for template-based structure prediction using ColabFold[74,75] to aid manual building of loop 1. Further refinement and validation were done as described above.

To make structural comparison possible, the pixel size of the apo GSEC1A structure (EMDB-3240)[28] was calibrated and the map was rescaled in the same way as described above. To avoid reinterpreting the map, the model (PDB: 5FN5) was refined into the rescaled map using real-space refinement in PHENIX and the original model was fitted onto the backbone of the refined model five amino acids at a time with an overlap of two amino acids to keep the sidechain orientations intact.

Figures of atomic models and cryo-EM maps were generated using UCSF ChimeraX[76] version 1.4.

### Bioinformatics analysis
APH-1 sequences were aligned using Clustal Omega[77] and coloured using Jalview[78] version 2.11.2.6 according to BLOSUM62 score with a 40% conservation threshold.

### Reporting summary
Further information on research design is available in the Nature Portfolio Reporting Summary linked to this article.

### Data availability
The cryo-EM density maps and atomic coordinates generated in this study have been deposited in the EMDB and the PDB under accession codes EMD-17112 (apo GSEC1B), EMD-17113 (GSEC1B-Aβ46), 8OQY (apo GSEC1B) and 8OQZ (GSEC1B-Aβ46). Publicly available PDB entries used in this study are available under accession codes 5FN2, 5FN5, 6IDF, 6IYC, 6LQG, 6LR4, 7C9I, 7Y5T. Source data are provided with this paper.

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

## Acknowledgements

We thank Annelore Stroobants for technical support, Dr. Marcus Fislage and Dr. Adam Schröfel for assistance with cryo-EM data collection. We would like to acknowledge the funding provided by the VIB international PhD program fellowship to I.O. and Fonds Wetenschappelijk Onderzoek (Grant Nos. G0H5916N, G054617N to R.G.E. and G0B2519N, G008023N to L.C.G).

## Author contributions

I.O. purified GSEC, reconstituted nanodiscs, performed cryo-EM analysis, analysed data, and wrote the original draft of the manuscript. M.B.H. purified GSEC, reconstituted nanodiscs, performed nanodisc-based and cell-based activity assays, and analysed data. S.L. performed cloning and generated the baculoviruses. L.C.G. and R.G.E. conceived, managed, and supervised the project, analysed data, reviewed, and edited the manuscript. I.O., L.C.G. and R.G.E. acquired funding.

## Competing interests

The authors declare no competing interests.
