## [Peer Review File · Nature Communications]

Apo and A β 46-bound γ -secretase structures provide insights into amyloid- β processing by the APH-1B isoformREVIEWER COMMENTS

Reviewer #1 (Remarks to the Author):

In this manuscript, the authors report the cryo-EM structure of GSEC complexes with and without the intermediate Ab46 substrate reconstituted in a nanodisc. They compared the determined structures in this study with the previously reported structure of GSEC complexes with APP or Notch substrates. Their findings are interesting and will be important to understand the mechanisms of APP cleavage that generates Ab peptides related AD, however, the issues listed below should be addressed before considering publication.

1. The authors generated an inactive GSEC1B enzyme by introducing mutation D275A. How do they confirm this mutation does not cause any conformational changes? Do they have any supporting experiments or references to support for the same conformation with this mutation?

3. For figures 2-5, I recommend labeling or highlighting the domains and residues that authors describe in the main manuscript. It is hard to follow what the authors describe in the figures. For examples,

- Fig. 3c: It seems zoomed region of Fig. 3b or Fig. 3a, but hard to understand how Fig.3c related to fig 3a and 3b.
- Fig 3a, b: in page 7 line 134, what are the three segments on the inner and outer membrane surfaces? Please indicate these three segments in the figure.
- Fig. 3: If authors indicate TM1,2,4,8, and 9 in Fig. 3c, then it would be much easier to follow.
- Fig. 4b: The color scheme does not agree between Fig.4a and Fig.4b. The helices labeled as 3 and 4 change their color from blue to pink and dark pink again with rotation in Fig. 4b. Do #3 and #4 helices represent the same helix during rotation? If yes, what does the color change mean? If not, authors should change their label.
- In page 8, line 154-155. Significant conformational difference between ... it would be beneficial to clearly mark the notable differences in figure.

4. This manuscript contains many typos and mismatched figure numbers. Since this work contains many proteins with isoforms, and multiple domains, typos make readers confused to follow their finding and claims. I recommend the authors read carefully again to correct them. These are examples that I found.

- Page 4, line 80. GSEC1A to GSEC1B
- Page 6, line 105 & 106, please correct Fig. 1c to Fig. 1e.
- Page 7, line 133, please correct Fig. 3b, c to Fig. 3e.

Also, authors use both APH-1B and APH1B, APH-1 and APH1, or PSEN-1 and PSEN1. I recommend using the consistent way to indicate protein.

Reviewer #2 (Remarks to the Author):

This manuscript presents a structural analysis of gamma-secretase in complex with Abeta 46. It is technically sound and well-written, highlighting intriguing details such as the impact of W165 on substrate stabilization. However, the overall organization and conceptualization of this paper may not sufficiently address specific research questions. While it encompasses a substantial amount of experimental work, it falls short of adequately addressing significant issues.

Two primary questions emerge from this manuscript that the authors aim to address:

1. The Structural-Functional Relationship between APH1a and APH1b: The data provided here is preliminary and inconclusive, as acknowledged by the authors in Page 8, lines 167 to 170. Technical variations in the structural differences between APH1 isoforms cannot be ruled out. To

answer this question comprehensively, the authors should conduct a comparative analysis of APH1a and APH1b complexes using the same methods to generate comparable data. Additionally, for Aph1B gamma-secretase, the use of DAPT or other inhibitors to stabilize the structure for visualizing TM2 and TM6 should be considered. To justify the mention of p88L effects on Aph1, the authors should analyze P88L-Aph1a and P88L-Aph1b complexes.

If this is the question authors wish to answer, they need to conduct a systematic comparison under the same lipid environment with more careful controls. The mentioning of p88L is not a justification itself for the Aph1 effects. If authors really want to mention this, they should resolve a P88L-Aph1a and P88L-Aph1b complexes.

2. The structural alignment between substrates and enzymes under endopeptidase or carboxyl-peptides mode of gamma-secretase. This is a very challenging question for any enzymologist. I appreciate the courage of the authors. But the data shown is not even patricianly addressing this. The similarity between the APh1b-abeta46 structure and the structure obtained in R Zhou, Science is concerning to me. As in Science paper, they have used an extremely artificial method to link the enzyme and substrate, which mutated APP and PS1 loop (important residues) into cysteine for oxidation crosslinking. As authors know very well, gamm-secretase is a very delicate enzyme maintaining a fragile carboxyl peptidase activity, which so many mutations could change the activities. The similarity of the structure presenting in this study, which is more biologically relevant, but still the PS1 NTF and CTF expressed separately, to the previous one. The Cryo-EM technology does not provide a dynamic range for observing changes, especially on a highly dynamic enzymatic complex.

So if authors really wish to answer this question "structural alignment between substrates and enzymes under endopeptidase or carboxyl-peptides mode of gamma-secretase". They will need to present a significant large dataset, including gamma-secretase-APP-CTF, gamma-secretase-49, gamma-secretase-46, gamma-secretase-43, gamma-secretase-40, and gamma-secretase-37.

In general, the manuscript would benefit from a more focused approach, addressing one of these questions in greater depth, given the limitations of the available data. Many of the conclusions drawn in the current manuscript remain speculative.

Additionally, there are several minor issues to address:

1. What is the difference between 8OYZ and 8OQZ PDB file mentioned in the text and table?
2. For Page 17, Line 370-376, in the field, we already know the importance of K28 for more than 13 years. It is not appropriate to cite the authors' own under review paper instead of the very detailed paper published in 2011. J Biol Chem. 2011 Nov 18;286(46):39804-12. I believe that authors should be aware of this publication as presenilin specialists.
3. Also for citation of specific FAD mutations, authors should cite the earliest functional analysis papers as well.
4. Figure 1d needs a series of western blots to show the the purification. Also authros need to explain the difference between the PSEN1-NTF intensity difference between the CHAPSO and NANOdisc. Also enzymatic assay should be done to screen all the abeta production, not only abeta 40. The schematic of Figure 1b is confusing, authors should provide real data to demonstrate the APh1a and 1b difference on the production line, instead of an ambiguous wave graph.

Reviewer #3 (Remarks to the Author):

Odorcic et al report the cryo-EM structures of the PS1/Aph1B g-secretase complex. Previous studies suggest that the Aph-1B complex has distinct activity for Ab production. The authors have solved the apo form with an overall resolution of 3.3 Å in nanodisc. Moreover, they show the structures of g-secretase bound with the substrate Ab46. They have found that the substrate binding stabilizes the complex and induce certain conformational changes. They compared both structures with previously published cryo-EM structures of the PS1/Aph1A g-secretase complex

and the complex bound with APP C83 substrates. The identification of interaction of Y115 in the loop1 with the substrate offers novel insights into the mechanism of γ -secretase processivity. This study further advances the understanding of γ -secretase proteolysis at the molecular and atomic level. This is an important work and will be of widespread interest to the readership of Nat. Communication

This manuscript would be strengthened if these questions could be addressed.

1. What's γ -secretase activity to produce A β 37, A β 38, A β 42 and A β 43 from the A β 46 substrate?
2. What is the spatial arrangement and distance of D275 and D385 in the Apo form structure? This will clarify whether this apo form is catalytically active.
3. How is this apo structure compared with the structure of γ -secretase complexed with L685458 (PMID: 33373587) that represents an active form of γ -secretase ?
4. Previous the cryo-EM structures of the γ -secretase complex were determined from proteins prepared in amphipols or detergents and current structures were obtained from nanodisc. Do these differences between the Aph1A and PS1/Aph1B reflect the nature of complexes or lipid environmental changes?
5. In Figure 6d, the authors proposed models of Ab43->Ab40 and Ab40->Ab37. Does a small subpopulation of Ab43->Ab40 particles exist?

We are thankful to the reviewers for reading our manuscript and providing valuable comments and suggestions. In response to their reviews, we revised our manuscript and below provide point-by-point responses to the reviewer's comments. As part of the revision, we performed additional analysis and experiments the results of which are presented in the new Supplementary Figures 1-3, Supplementary Table 2, and modified Figure 1. In the revised manuscript the modifications are highlighted in yellow.

Reviewer #1 (Remarks to the Author):

In this manuscript, the authors report the cryo-EM structure of GSEC complexes with and without the intermediate A β 46 substrate reconstituted in a nanodisc. They compared the determined structures in this study with the previously reported structure of GSEC complexes with APP or Notch substrates. Their findings are interesting and will be important to understand the mechanisms of APP cleavage that generates Ab peptides related AD, however, the issues listed below should be addressed before considering publication.

1. The authors generated an inactive GSEC1B enzyme by introducing mutation D275A. How do they confirm this mutation does not cause any conformational changes? Do they have any supporting experiments or references to support for the same conformation with this mutation?

We have noticed that Reviewer #1 and Reviewer #3 refer to D275A instead of D257A. We have found a typo in the first mention of this mutant in the paper (page 5, line 104) and apologize for the confusion. In the revised manuscript, "PSEN1 D275A" has been corrected to "PSEN1 D257A".

Response: We thank the reviewer for bringing up this important point which was not discussed in our initial submission. Mutagenesis of catalytic residues that inactivate the protease is a common strategy used in structural analysis of complexes of proteases with their substrates. This strategy was previously used by Zhou, Yang et al., 2019 and Yang et al., 2021 to solve the structure of GSEC1A-APP_{C83} and the GSEC1A-Notch complexes. In these reports, D385, the other aspartate of the catalytic dyad in PSEN1 (D257/D385) was mutated to Ala.

To investigate potential conformational changes caused by the mutation of one (or the other) catalytic Asp, we compared the structures of the active sites in the following structures: mutant GSEC1B (PSEN1^{D257A}) in complex with A β 46, mutant GSEC1A (PSEN1^{D385A}) in complex with Notch (**Supplementary Figure 3a**) or APP_{C83} (**Supplementary Figure 3b**), and WT GSEC1A in complex with a transition-state analogue (**Supplementary Figure 3c**). Our analysis shows that the active site structures in the four GSEC complexes are very similar, and therefore independent of the D385A and D257A mutations.

It is also worth noting that the D257A mutation blocks PSEN1 endoproteolysis and produces an inactive GSEC complex containing full-length PSEN1. Therefore, in addition to introducing the D257A mutation, we expressed the mutant (D257A) PSEN1 as N- and C-terminal fragments.

Action: We have added Supplementary Figure 3 which shows the structural alignment of PSEN1^{D257A} with PSEN1^{WT} and PSEN1^{D385A} from inhibitor-bound and substrate-bound structures. On page 9, line 193, we have added the following text: "The overall structure is remarkably similar to the GSEC1A in complex with Notch, APP_{C83} or the transition state analogue inhibitor L-685,458, which induces a substrate-bound-like conformation. In particular, the structure of the catalytic dyad in PSEN1^{D257A} of GSEC1B-A β 46, PSEN1^{D385A} of GSEC1A-APP_{C83}, and PSEN1^{WT} of GSEC1A-Inhibitor (L-685,458) is virtually identical (**Supplementary Figure 3, Supplementary Table 2**) indicating that neither D257A nor D385A mutations alter the active site structure."

2. For figures 2-5, I recommend labeling or highlighting the domains and residues that authors describe in the main manuscript. It is hard to follow what the authors describe in the figures.

• For examples, Fig. 3c: It seems zoomed region of Fig. 3b or Fig. 3a, but hard to understand how Fig.3c related to fig 3a and 3b.

Response: We thank the reviewer for the suggestions to improve the comprehensibility of the figures. We have adjusted the figures as described below.

Action: In Figure 2, the labels of GSEC subunits have been moved and enlarged to improve readability. The inset panels in Figure 2b have been connected to the corresponding boxes on the cryo-EM map. In Figures 3-5, the colour code of the labels of GSEC subunits has been moved below the corresponding panels, and the labels for TM helices have been changed to colour-coded circled numbers. In Figure 2, panels b and c are now linked to the corresponding colour-coded boxes in Figure 2a, which have been added.

3. Fig 3a, b: in page 7 line 134, what are the three segments on the inner and outer membrane surfaces? Please indicate these three segments in the figure.

Response: The three segments are those indicated with dashed boxes in Figure 3b, e. We have edited the text and the figure to make this clearer.

Action: On page 7, line 141 the part of the sentence "... the backbones of the isoforms diverged locally at three segments on the inner (TM3-4) and outer (TM2-3 and TM6-7) membrane surfaces, with changes mapping to the PSEN1/APH-1 interface (Figure 3a, b)." has been changed to "... the backbones of the isoforms diverged locally at three segments on the intracellular (TM3-4) and extracellular (TM2-3 and TM6-7) membrane surfaces, with changes mapping to the PSEN1/APH-1 interface (**Figure 3a, b**, red dashed boxes)". In Figure 3b, e, the three segments are now indicated with a thicker red dashed box. Additionally, to improve clarity in Figure 3e, the colours representing the conservation of aligned residues have been inverted.

4. Fig. 3: If authors indicate TM1,2,4,8, and 9 in Fig. 3c, then it would be much easier to follow.

Response: All TM helices are now labelled in Figure 3. PSEN1 TMs 1, 2 and 4 are not shown for clarity; however, this figure has been edited as described above. The boxes indicating zoomed regions were added along with an indication of rotation angles which should facilitate reader's navigation in the figure.

5. Fig. 4b: The color scheme does not agree between Fig.4a and Fig.4b. The helices labeled as 3 and 4 change their color from blue to pink and dark pink again with rotation in Fig. 4b. Do #3 and #4 helices represent the same helix during rotation? If yes, what does the color change mean? If not, authors should change their label.

Response: Figure 4 was edited to address the ambiguities brought up by the reviewer.

Action: In Figure 4 panels a and b have been merged and the labelling has been improved as described above. APH-1 TMs 3 and 4 have been added to the second panel.

6. In page 8, line 154-155. Significant conformational difference between ... it would be beneficial to clearly mark the notable differences in figure.

Response: We agree with the reviewer and have therefore created the Supplementary Figure 2 to show the conformational differences in detail.

Action: Supplementary Figure 2 was added to the revised manuscript. On page 8, line 161, the reference to this figure indicates: "Significant conformational differences between GSEC1A and GSEC1B structures in apo states are observed in the catalytic PSEN1 subunit (**Supplementary Figure 2**)."

7. This manuscript contains many typos and mismatched figure numbers. Since this work contains many proteins with isoforms, and multiple domains, typos make readers confused to follow their finding and claims. I recommend the authors read carefully again to correct them. These are examples that I found.

Response: We have checked the manuscript for typos and fixed the ones we found as described below.

8. Page 4, line 80. GSEC1A to GSEC1B

The typo has been corrected.

Action: On page 5, line 81, PSEN1/APH-1B has been corrected to PSEN1/APH-1A.

9. Page 6, line 105 & 106, please correct Fig. 1c to Fig. 1e.

We double checked the label and found that it was correct in the manuscript as submitted.

10 Page 7, line 133, please correct Fig. 3b, c to Fig. 3e.

We double-checked the label and found that it was correct in the manuscript as submitted .

11. Also, authors use both APH-1B and APH1B, APH-1 and APH1, or PSEN-1 and PSEN1. I recommend using the consistent way to indicate protein.

APH-1B is the name of the human gene that encodes the APH-1B protein. In the original version the gene name was not italicized which has been corrected in the revised manuscript. The instance of PSEN1 is a typo and has been fixed in the revised version.

Action: On page 3, line 53, “mouse” has been added in front of *APH-1B* to make it clear that it refers to the mouse gene. On page 3, line 55, APH-1B has been italicised because it refers to the human gene. On page 5, line 82 PSEN-1/APH-1 has been changed to PSEN1/APH-1. On page 7, line 146, APH1-A has been changed to APH-1A.

Reviewer #2 (Remarks to the Author):

1. This manuscript presents a structural analysis of gamma-secretase in complex with Aβ46. It is technically sound and well-written, highlighting intriguing details such as the impact of W165 on substrate stabilization. However, the overall organization and conceptualization of this paper may not sufficiently address specific research questions. While it encompasses a substantial amount of experimental work, it falls short of adequately addressing significant issues.

Response: We thank the reviewer for acknowledging that our studies are technically sound. It should be taken into consideration, that structural studies of GSEC complexes remain challenging, limiting our possibilities of generating structures of many complexes.

Although our work does not answer all the possible scientific questions related to the modulation of GSEC activity by APH-1 isomorphism and the details of the substrate processivity mechanism, the presented data do make significant advances toward answering both questions. These include the following: 1) our manuscript is the first report of GSEC structure in apo and substrate-bound form determined under the same conditions, and thus enables direct comparison of these states; 2) Our structures define ‘regions of interest’ in APH-1 and constraints on the extent of conformational changes associated with allosteric modulation of GSEC by APH-1 isoforms. Follow-up studies, as indicated in the following point, are required to address the significance of the observed conformational changes; 3) We report the first structure of GSEC in complex with a substrate in the

absence of E-S crosslinking, and 4) this is the first structure of GSEC in complex with an intermediate A β substrate.

2. Two primary questions emerge from this manuscript that the authors aim to address:

The Structural-Functional Relationship between APH-1a and APH-1b: The data provided here is preliminary and inconclusive, as acknowledged by the authors in Page 8, lines 167 to 170. Technical variations in the structural differences between APH-1 isoforms cannot be ruled out. To answer this question comprehensively, the authors should conduct a comparative analysis of APH-1a and APH-1b complexes using the same methods to generate comparable data.

Response:

The availability of structural data for the GSEC1A in complex with A β 46 in lipid nanodiscs would be valuable in drawing conclusions, but, technically, the structure determination of such a complex would not be feasible within a reasonable timeframe. Our structures are informative. They define ‘regions of interest’ in APH-1 and constraints on the extent of conformational changes associated with allosteric modulation of GSEC by APH-1 isoforms. Thus, our structural data do provide a strong basis for testable functional hypotheses addressing the structural bases of the differential roles of APH-1 isoforms. The influence of specific APH-1 regions on the GSEC processivity can be addressed by APH-1A/B chimeras limited to the regions within which structural changes were observed. This relevant analysis will be our future research direction for the investigation of the allosteric role of APH-1 subunit isomorphism. Furthermore, and importantly, the observed structural differences are consistent with structural predictions made by AlphaFold2, suggesting that they are isoform-driven.

Action: On page 18, line 403, the following sentence has been added: “Regarding the allosteric-like role of APH-1, our structural data define ‘regions of interest’ in APH-1 and thus provide a basis for the design and functional analysis of APH-1A/B chimeras, restricted to the regions where structural differences are observed.”.

3. Additionally, for APH-1B gamma-secretase, the use of DAPT or other inhibitors to stabilize the structure for visualizing TM2 and TM6 should be considered.

Response: It is true that in the absence of substrates these two helices are dynamic and as such cannot be visualized by cryo-EM, and that GSEC inhibitors do stabilise the otherwise disordered helices TM2 and TM6. Binding of DAPT to GSEC will stabilise the enzyme in the inhibited state, as reported in Bai et al., 2015. The analysis of GSEC inhibition is not within the scope of the present study. However, it is worth noting that in the GSEC1A structure solved in the presence of DAPT, the conformation of the PSEN1 subunit is very similar to the substrate-bound conformation; thus, the GSEC1B-DAPT structure will likely be similar to the A β 46-bound structure.

4. To justify the mention of p88L effects on APH-1, the authors should analyze P88L-APH-1a and P88L-APH-1b complexes.

If this is the question authors wish to answer, they need to conduct a systematic comparison under the same lipid environment with more careful controls. The mentioning of p88L is not a justification itself for the APH-1 effects. If authors really want to mention this, they should resolve a P88L-APH-1a and P88L-APH-1b complexes.

Response: We thank the reviewer for raising this specific point and allowing us to clarify it. We have used the effects of the P88L mutation on the length of A β products as evidence for the importance of TM1^{PSEN1} dynamics for GSEC sequential catalysis. Considering that (1) TM1^{PSEN1}, TM8^{PSEN1} and TM3-TM4^{APH-1} undergo concerted motion between the apo and substrate-bound states, including a hinge motion at P88; and (2) that the removal of this hinge by the P88L mutation in PSEN1, reduces the efficiency of GSEC sequential catalysis more than any other described mutation, we suggest that

APH-1 isoforms might modulate GSEC sequential catalysis by affecting the dynamics of TM1^{PSEN1} via concerted motions mentioned above (1). We have edited the revised manuscript to bring this point into consideration.

Action: On page 10, line 221, the text “FAD-causing P88L mutation which interferes with TM1^{PSEN1} dynamics strongly impairs processivity^{33,34}. In a similar manner, APH-1 isoforms might change the environment of TM1^{PSEN1} and modulate GSEC processivity” was replaced with “Interestingly, the FAD-causing P88L mutation strongly impairs processivity and causes release of Aβ45 and longer species *in vitro*^{33,34}. Since Leu, a helix-promoting residue, impairs the hinge property enabled by Pro, a kink-forming residue, this suggests that the described motion of TM1^{PSEN1} is necessary for sequential catalysis”. On page 15, line 352, the text “This proposal is supported by the effect of AD-linked P88L mutation in TM1^{PSEN1}³⁴, which replaces the helix-breaking Pro with a helix-forming Leu. This aggressive pathogenic PSEN1 variant exerts marked destabilising effects on GSEC-Aβ interactions¹⁸ and thereby promotes the release of partially digested Aβ peptides, including very long Aβ45/46 peptides *in vitro*³³. Similar to this pathogenic mutation, differences in the environment surrounding TM1^{PSEN1} provided by APH-1A or APH-1B isoforms may change its dynamics and, consequently, alter GSEC processivity.” was replaced with “Interestingly, the FAD-causing P88L mutation strongly impairs processivity and causes release of Aβ45 and longer Aβ species *in vitro*^{33,34}. Because Leu, a helix-promoting residue, removes the hinge property enabled by Pro, a kink-forming residue, the effects of the pathogenic mutation suggest that the described motion of TM1^{PSEN1} is necessary for sequential catalysis.”.

5. The structural alignment between substrates and enzymes under endopeptidase or carboxyl-peptides mode of gamma-secretase. This is a very challenging question for any enzymologist. I appreciate the courage of the authors. But the data shown is not even patricianly addressing this. The similarity between the APH-1b- β 46 structure and the structure obtained in R Zhou, Science is concerning to me. As in Science paper, they have used an extremely artificial method to link the enzyme and substrate, which mutated APP and PS1 loop (important residues) into cysteine for oxidation crosslinking. As authors know very well, gamm-secretase is a very delicate enzyme maintaining a fragile carboxyl peptidase activity, which so many mutations could change the activities. The similarity of the structure presenting in this study, which is more biologically relevant, but still the PS1 NTF and CTF expressed separately, to the previous one. The Cryo-EM technology does not provide a dynamic range for observing changes, especially on a highly dynamic enzymatic complex.

So if authors really wish to answer this question "structural alignment between substrates and enzymes under endopeptidase or carboxyl-peptides mode of gamma-secretase". They will need to present a significant large dataset, including gamma-secretase-APP-CTF, gamma-secretase-49, gamma-secretase-46, gamma-secretase-43, gamma-secretase-40, and gamma-secretase-37.

Response: We thank the reviewer for considering our structures more biologically relevant, we do agree that GSEC processivity is fragile and sensitive to environment and mutations. This is why we solved GSEC apo and substrate-bound structures under conditions that mimic, as much as possible, the native environment of the membrane (lipid nanodiscs), and facilitate normal E-S interaction (no cross-linking). Importantly, we focussed our analysis on the GSEC-Aβ46 complex with the aim of providing insights on the structural basis of the (γ) E-S interactions and stability, given their critical role in defining the efficiency of the carboxypeptidase activity (processivity) (Szaruga et al, 2017).

We fully agree that the elucidation of all (γ) E-S co-structures, involved in the different stages of the GSEC-mediated APP cleavage, will be of fundamental and translational value, as they may provide key insights into the recognition and stabilisation of the different Aβ peptides during sequential proteolysis; thus, providing insights into how substrate shortening progressively promotes E-S release. However, the challenge of achieving these structures is significant, given the increasingly higher

aggregation propensities of A β peptides longer than A β 42, and the decreasing affinity of shorter A β substrates (shorter than A β 46, Szaruga et al, 2017). The indirect relationship between the solubility and aggregation propensity of A β peptides, and their relative affinities for the enzyme complicates the reconstitution of stable E-S complexes, which has a direct impact on the amount of required cryo-EM data and image processing efforts. We will work on this in the future, but this work is likely to take years to complete. The challenge of studying GSEC structures should not be underestimated; for example solving GSEC-APP_{C83} or GSEC-Notch structures required crosslinking of E-S complexes and took a leading structural group four years to achieve (judging from the time between relevant publications).

Action: To better explain this, we have added the following text to the discussion: on page 18, line 407 the text “To fully understand the sequential mechanism, the elucidation of the (γ) E-S co-structures, involved in the different stages of the GSEC-mediated APP cleavage is required, as this will provide key insights into the recognition and stabilization of A β peptides during sequential proteolysis. The challenge of achieving these structures is however significant, given the increasingly higher aggregation propensities of A β peptides longer than A β 42, and the decreasing affinity of peptides shorter than A β 46¹⁸.” has been added.

Response: When it comes to cryo-EM, unlike X-ray crystallography, this technique can visualize protein dynamics by resolving multiple discrete conformations present in the assembly or by using recently developed machine-learning techniques that can resolve the continuum of conformational states. Using 3D classification, we were able to, for example, separate GSECs containing resolved A β from the rest (Extended Data Figure 4 - the last 3D classification). With regard to the conformational dynamics of the substrate, we have tried various approaches to resolve the separate structures, however, with no success. We suspect that multiple substrate positions cannot be resolved by cryo-EM due to the very small moving mass involved, therefore we are limited to observation of substrate density which is a superposition of multiple conformations.

Action: To better explain this, we have added the following text on page 11, line 228 “We have attempted local 3D classification as well as employed various heterogeneity analyses on the substrate and the surrounding area, but nothing yielded a higher resolution reconstruction.”.

6. In general, the manuscript would benefit from a more focused approach, addressing one of these questions in greater depth, given the limitations of the available data. Many of the conclusions drawn in the current manuscript remain speculative.

Response: We have addressed the specific points related to the focus and conclusions in the answers to the points above.

Additionally, there are several minor issues to address:

7. What is the difference between 8OYZ and 8OQZ PDB file mentioned in the text and table?

Response: 8OQY is the structure of apo GSEC1B and 8OQZ is the structure of GSEC1B in complex with A β 46. This is described in the Data Availability section on page 31, line 684.

8. For Page 17, Line 370-376, in the field, we already know the importance of K28 for more than 13 years. It is not appropriate to cite the authors' own under review paper instead of the very detailed paper published in 2011. J Biol Chem. 2011 Nov 18;286(46):39804-12. I believe that authors should be aware of this publication as presenilin specialists.

Response: We thank the reviewer for raising this point, we have modified the manuscript accordingly.

Action: On page 18, line 395, the sentence “This mechanism is also supported by our recent data demonstrating that charged/polar residues in the ectodomain of the APP limit substrate threading and

promote product release⁴⁹.” was changed to “This mechanism is also supported by reports in literature showing that the K28A mutation causes release of very short A β species^{53,54}, as well as our recent data demonstrating that charged/polar residues in the ectodomain of the APP limit substrate threading and promote product release. Sequential proteolysis of the amyloid precursor protein (APP) by γ -secretases generates amyloid- β (A β) peptides and defines the proportion of short-to-long A β peptides, which is tightly connected to Alzheimer's disease (AD) pathogenesis. Here, we study the mechanism that controls substrate processing by γ -secretases and A β peptide length. We found that polar interactions established by the APPC99 ectodomain (ECD), involving but not limited to its juxtamembrane region, restrain both the extent and degree of γ -secretases processive cleavage by destabilizing enzyme-substrate interactions. We show that increasing hydrophobicity, via mutation or ligand binding, at APPC99 -ECD attenuates substrate-driven product release and rescues the effects of Alzheimer's disease-associated pathogenic γ -secretase and APP variants on A β length. In addition, our study reveals that APPC99 -ECD facilitates the paradoxical production of longer A β s caused by some γ -secretase inhibitors, which act as high-affinity competitors of the substrate. These findings assign a pivotal role to the substrate ECD in the sequential proteolysis by γ -secretases and suggest it as a sweet spot for the potential design of APP-targeting compounds selectively promoting its processing by these enzymes. APP substrate ectodomain defines amyloid- β peptide length by restraining γ -secretase processivity and facilitating product release”.

9. Also for citation of specific FAD mutations, authors should cite the earliest functional analysis papers as well.

Response: The appropriate citations have been added:

Campion, D. *et al.* Mutations of the presenilin I gene in families with early-onset Alzheimer's disease. *Hum Mol Genet* **4**, 2373–2377 (1995)

Murayama, O. *et al.* Enhancement of amyloid beta 42 secretion by 28 different presenilin 1 mutations of familial Alzheimer's disease. *Neurosci Lett* **265**, 61–63 (1999)

De Jonghe, C. *et al.* Aberrant splicing in the presenilin-1 intron 4 mutation causes presenile Alzheimer's disease by increased A β 42 secretion. *Hum Mol Genet* **8**, 1529–1540 (1999)

Sun, L., Zhou, R., Yang, G. & Shi, Y. Analysis of 138 pathogenic mutations in presenilin-1 on the in vitro production of A β 42 and A β 40 peptides by γ -secretase. *Proc Natl Acad Sci U S A* **114**, E476–E485 (2017)

10. Figure 1d needs a series of western blots to show the purification.

Response: We have performed western blots and the results are presented in **Supplementary Figure 1b**.

11. Also authors need to explain the difference between the PSEN1-NTF intensity difference between the CHAPSO and NANODisc.

Response: The intensity difference stems from contaminants that are removed during nanodisc reconstitution. As shown by the western blot (**Supplementary Figure 1b**), the band intensity of this, and other GSEC components is consistent between the conditions.

12. Also enzymatic assay should be done to screen all the abeta production, not only abeta 40.

Response: We have optimised the nanodisc-based activity assays for GSEC1A and GSEC1B and performed ELISA measurements for A β 37, A β 38, A β 40, A β 42 and A β 43. *De novo* Ab production was quantified using the reconstituted inactive enzyme as reference.

Action: Figure 1 and Supplementary Figure 1 have been modified to represent the *de novo* Ab production data and text has been updated to reflect the changes.

On page 6, line 107 the text “Incubation of A β 46 with the reconstituted wild type GSEC1B complex resulted in generation of A β 40 (**Figure 1e**), while the reconstituted GSEC1B^{PSEN1 D257A} mutant did not hydrolyse A β 46 (**Figure 1e**). However, the latter formed stable E-S complexes (see below).” was changed to “Incubation of A β 46 with the reconstituted wild-type (WT) GSEC1A and GSEC1B complexes resulted in generation of shorter A β products (**Figure 1e** and **Supplementary Figure 1c**). Strikingly, *de novo* A β production revealed that the processivity difference between the isoforms in nanodisc conditions is even greater than previously reported for detergent conditions¹¹. The reconstituted inactive GSEC1B^{PSEN1 D257A} mutant did not hydrolyse A β 46, however, it formed stable E-S complexes in the absence of covalent cross-linking (see below).”.

13.The schematic of Figure 1b is confusing, authors should provide real data to demonstrate the APH-1a and 1b difference on the production line, instead of an ambiguous wave graph.

Response: Our original intent with this panel was to show the difference in processivity schematically and in a comprehensible manner to the general audience. Following the request of the reviewer, we have adjusted the figure to represent the relative amount of A β released by each isoform as reported in Acx et al, 2013. Additionally, we have performed activity assays with GSEC1A and GSEC1B in nanodiscs as described above and this data is displayed in **Figure 1e**.

Reviewer #3 (Remarks to the Author):

Odorcic et al report the cryo-EM structures of the PS1/APH-1B g-secretase complex. Previous studies suggest that the Aph-1B complex has distinct activity for Ab production. The authors have solved the apo form with an overall resolution of 3.3 Å in nanodisc. Moreover, they show the structures of g-secretase bound with the substrate A β 46 . They have found that the substrate binding stabilizes the complex and induce certain conformational changes. They compared both structures with previously published cryo-EM structures of the PS1/APH-1A g-secretase complex and the complex bound with APP C83 substrates. The identification of interaction of Y115 in the loop1 with the substrate offers novel insights into the mechanism of g-secretase processivity. This study further advances the understanding of g-secretase proteolysis at the molecular and atomic level. This is an important work and will be of widespread interest to the readership of Nat. Communication

This manuscript would be strengthened if these questions could be addressed.

1. What’s g-secretase activity to produce A β 37, A β 38, A β 42 and A β 43 from the A β 46 substrate?

Response: A similar question was raised by Reviewer #2. We have now measured the *de novo* products generated from A β 46 by GSEC1A and GSEC1B in nanodiscs. We have optimised the nanodisc-based activity assays for GSEC1A and GSEC1B and performed ELISA measurements for A β 37, A β 38, A β 40, A β 42, and A β 43. *De novo* Ab production was quantified using the reconstituted inactive enzyme as a reference.

Action: Figure 1 and Supplementary Figure 1 have been modified to represent the *de novo* Ab production data and text has been updated to reflect the changes.

On page 6, line 107 the text “Incubation of A β 46 with the reconstituted wild type GSEC1B complex resulted in generation of A β 40 (**Figure 1e**), while the reconstituted GSEC1B^{PSEN1 D257A} mutant did not hydrolyse A β 46 (**Figure 1e**). However, the latter formed stable E-S complexes (see below).” was changed to “Incubation of A β 46 with the reconstituted wild-type (WT) GSEC1A and GSEC1B

complexes resulted in generation of shorter A β products (**Figure 1e** and **Supplementary Figure 1c**). Strikingly, *de novo* A β production revealed that the processivity difference between the isoforms in nanodisc conditions is even greater than previously reported for detergent conditions¹¹. The reconstituted inactive GSEC1B^{PSEN1 D257A} mutant did not hydrolyse A β 46, however, it formed stable E-S complexes in the absence of covalent cross-linking (see below).”.

2. What is the spatial arrangement and distance of D275 and D385 in the Apo form structure? This will clarify whether this apo form is catalytically active.

Response: In GSEC1B apo structure D257 is not resolved because the density of the corresponding helix (TM6) is flexible and could not be modelled starting from exactly residue D257; therefore, this question cannot be answered based on our experimental data only. However, if we take into consideration the (only) other apo structure of GSEC (GSEC1A structure solved in amphipols, **Figure 3** and **Supplementary Figure 2**), we observe that TM6 and TM7 in these structures very closely align, suggesting that the D257-D385 distances are similar. To provide quantitative insights into this point, we measured the D257-D385 distances in GSEC apo, substrate- and inhibitor- bound structures and listed them in a new Supplementary Table 2. It is important to note, however, that the conformational changes upon substrate binding involve large rearrangements of TM6 and hence the catalytic aspartates are not assembled in proteolytically active geometry in apo structures.

Reviewer #3 may find of interest the first point raised by Reviewer #1 and the associated response.

Actions: Supplementary Table 2 has been added in the revised manuscript.

3. How is this apo structure compared with the structure of g-secretase complexed with L685458 (PMID: 33373587) that represents an active form of g-secretase?

Response: The GSEC1B apo structure significantly differs from that of GSEC1A in complex with the transition state analogue inhibitor L-685,458 (**Supplementary Figure 3d**). The available structural data for GSEC indicate that the structures of GSEC with inhibitors have a PSEN1 conformation similar to the substrate-bound structures, rather than to apo GSEC structures. In fact, the comparison of the substrate- and L-685,458- bound structures (**Supplementary Figure 3c**) revealed that the active conformation (GSEC- L-685,458 complex) is very similar to the GSEC1A in complex with APP_{C83}, except for PSEN1 loop 1, which we have described in the manuscript (**Supplementary Figure 3b**).

Action: In the revised manuscript, we added Supplementary Figure 3d which shows the alignment between GSEC1B apo structure and GSEC1A in complex with L-685,458. We have expanded the text to include the comparison of GSEC1B with inhibitor-bound GSEC1A. (This is also described in the response to the similar Reviewer #1's 1st point). We have added Supplementary Figure 3 which shows the structural alignment of PSEN1^{D257A} with PSEN1^{WT} and PSEN1^{D385A} from inhibitor-bound and substrate-bound structures. On page 9, line 193, we have added the following text: “The overall structure is remarkably similar to the GSEC1A in complex with Notch, APP_{C83} or the transition state analogue inhibitor L-685,458, which induces a substrate-bound-like conformation. In particular, the structure of the catalytic dyad in PSEN1^{D257A} of GSEC1B-A β 46, PSEN1^{D385A} of GSEC1A-APP_{C83}, and PSEN1^{WT} of GSEC1A-Inhibitor (L-685,458) is virtually identical (**Supplementary Figure 3, Supplementary Table 2**) indicating that neither D257A nor D385A mutations alter the active site structure.”.

4. Previous the cryo-EM structures of the g-secretase complex were determined from proteins prepared in amphipols or detergents and current structures were obtained from nanodisc. Do these differences between the APH-1A and PS1/APH-1B reflect the nature of complexes or lipid environmental changes?

Response: (A similar point was raised by Reviewer #2 in their 2d comment). The observed structural differences in APH-1A/B are consistent with predictions made by AlphaFold2, suggesting that the observed differences are isoform-driven. However, we cannot draw definitive conclusions, as discussed in the manuscript. Our structures are informative. They define ‘regions of interest’ in APH-1 and constraints on the extent of conformational changes associated with allosteric modulation of GSEC by APH-1 isoforms. Thus, our structural data do provide a strong basis for testable functional hypotheses addressing the structural bases of the differential roles of APH-1 isoforms. The influence of specific APH-1 regions on the GSEC processivity can be addressed by APH-1A/B chimeras limited to the regions within which structural changes were observed. This relevant analysis will be our future research direction for the investigation of the allosteric role of APH-1 subunit isomorphism. Furthermore, and importantly, the observed structural differences are consistent with structural predictions made by AlphaFold2, suggesting that they are isoform-driven.

Action: On page 18, line 403, the following sentence has been added: “Regarding the allosteric-like role of APH-1, our structural data define ‘regions of interest’ in APH-1 and thus provide a basis for the design and functional analysis of APH-1A/B chimeras, restricted to the regions where structural differences are observed.”.

5. In Figure 6d, the authors proposed models of Ab43->Ab40 and Ab40->Ab37. Does a small subpopulation of Ab43->Ab40 particles exist?

Response: We are unsure what the reviewer means exactly with this question.

If the Reviewer refers to the option that some A β 46 peptides are bound in register for the A β 43 \rightarrow A β 40 reaction, we have considered that option and discussed it in the manuscript. As mentioned above in response to Reviewer #2, we have not been able to separate the distinct A β 46 conformations/positions. We refer to this point in the manuscript on page 11, line 249: “We have attempted local 3D classification as well as well as employed various heterogeneity analyses on the substrate and the surrounding area, but nothing yielded a higher resolution reconstruction.”

If the Reviewer refers to the possibility that A β 46 might be contaminated with A β 43, we cannot rule it out completely, however, according to the manufacturer, the peptide is $\geq 95\%$ pure, as determined by mass spectrometry (see below, provided by the manufacturer; theoretical m/z values have been indicated). In addition, our analysis of the A β 46 peptide in urea gels show a major band at the expected peptide mobility (see below) consistent with high purity.

Mass spectrometry analysis of A β 46.

Urea SDS-PAGE analysis of A β 46.

REVIEWERS' COMMENTS

Reviewer #1 (Remarks to the Author):

The authors have addressed all suggestions, and I don't think this manuscript needs to go another round of reviews.

Reviewer #2 (Remarks to the Author):

Authors answered my technical questions carefully.

Reviewer #3 (Remarks to the Author):

The authors have reasonably addressed and discussed issues I raised. I have no major issues. Minor issue.

Is this true that both forms of GSEC can generate Ab42 and Ab38 from Ab46 substrate in Fig 1e? If yes, the implication of processivity of g-secretase regarding Ab49 and Ab48 lines should be discussed.

REVISION 2

Reviewer #1 (Remarks to the Author):

The authors have addressed all suggestions, and I don't think this manuscript needs to go another round of reviews.

Reviewer #2 (Remarks to the Author):

Authors answered my technical questions carefully.

Reviewer #3 (Remarks to the Author):

The authors have reasonably addressed and discussed issues I raised. I have no major issues. Minor issue.

Is this true that both forms of GSEC can generate Ab42 and Ab38 from Ab46 substrate in Fig 1e? If yes, the implication of processivity of γ -secretase regarding Ab49 and Ab48 lines should be discussed.

Response: This is correct and has been previously reported by Matsumura et al, JBC 2014; 21;289(8):5109-21. Matsumura *et al* comprehensively characterized the processing of A β peptides by γ -secretases. Their analysis shows that A β 46 is primarily converted to A β 43, but A β 42 and A β 38 peptides are also generated albeit to a lesser extent.

Action: We have modified figure 1b to indicate in more detail the processing of A β peptides by γ -secretase. Moreover, we added a note in the results section (page 6 line 111) of the revised manuscript which reads as follows:

Incubation of A β 46 with the reconstituted wild-type (WT) GSEC1A and GSEC1B complexes resulted in generation of shorter A β products, including A β 43, A β 40 and A β 37 as well as A β 42 and A β 38 (**Figure 1e** and **Supplementary Figure 1c**). We note that A β 46 is mainly converted into A β 43, A β 40 and A β 37, but its processing to A β 42 and A β 38 also occurs albeit to a lesser extent².